# Dengue virus infection induces selective expansion of Vγ4 and Vγ6TCR γδ T cells in the small intestine and a cytokine storm driving vascular leakage in mice

Takeshi Kurosu[1]*, Daisuke Okuzaki[2]*, Yusuke Sakai[3], Mohamad Al Kadi[2], Supranee Phanthanawiboon[1¤], Yasusi Ami[4], Masayuki Shimojima[1], Tomoki Yoshikawa[1], Shuetsu Fukushi[1], Noriyo Nagata[3], Tadaki Suzuki[3], Daisuke Kamimura[5], Masaaki Murakami[5,6,7,8], Hideki Ebihara[1], Masayuki Saijo[1]

1 Department of Virology I, National Institute of Infectious Diseases, Tokyo, Japan, 2 Laboratory of Human Immunology (Single Cell Genomics), WPI Immunology Research Center, Osaka University, Suita, Osaka, Japan, 3 Department of Pathology, National Institute of Infectious Diseases, Tokyo, Japan, 4 Management Department of Biosafety, Laboratory Animal, and Pathogen Bank, National Institute of Infectious Diseases, Tokyo, Japan, 5 Division of Molecular Psychoimmunology, Institute for Genetic Medicine and Graduate School of Medicine, Hokkaido University, Sapporo, Japan, 6 Team of Quantumimmunology, Institute for Quantum Life Science, National Institute for Quantum and Radiological Science and Technology (QST), Chiba, Japan, 7 Division of Molecular Neuroimmunology, Department of Homeostatic Regulation, National Institute for Physiological Sciences, National Institutes of Natural Sciences, Aichi, Japan, 8 Institute for Vaccine Research and Development (HU-IVReD), Hokkaido University, Sapporo, Japan

¤ Current address: S.P., Department of Microbiology, Faculty of Medicine, Khon Kaen, Thailand
* kurosu@niid.go.jp (TK); dokuzaki@biken.osaka-u.ac.jp (DO)

**Data Availability Statement:** All relevant data are within the manuscript and its Supporting Information files.

## Abstract

Dengue is a major health problem in tropical and subtropical regions. Some patients develop a severe form of dengue, called dengue hemorrhagic fever, which can be fatal. Severe dengue is associated with a transient increase in vascular permeability. A cytokine storm is thought to be the cause of the vascular leakage. Although there are various research reports on the pathogenic mechanism, the complete pathological process remains poorly understood. We previously reported that dengue virus (DENV) type 3 P12/08 strain caused a lethal systemic infection and severe vascular leakage in interferon (IFN)-α/β and γ receptor knockout mice (IFN-α/β/γRKO mice), and that blockade of TNF-α signaling protected mice. Here, we performed transcriptome analysis of liver and small intestine samples collected chronologically from P12/08-infected IFN-α/β/γRKO mice in the presence/absence of blockade of TNF-α signaling and evaluated the cytokine and effector-level events. Blockade of TNF-α signaling mainly protected the small intestine but not the liver. Infection induced the selective expansion of IL-17A-producing Vγ4 and Vγ6 T cell receptor (TCR) γδ T cells in the small intestine, and IL-17A, together with TNF-α, played a critical role in the transition to severe disease via the induction of inflammatory cytokines such as TNF-α, IL-1β, and particularly the excess production of IL-6. Infection also induced the infiltration of neutrophils, as well as neutrophil collagenase/matrix metalloprotease 8 production. Blockade of IL-17A signaling reduced mortality and suppressed the expression of most of these cytokines, including TNF-α, indicating that IL-17A and TNF-α synergistically enhance cytokine expression.

**Funding:** This work was supported by a grant-in-aid from the Ministry of Education, Culture, Sports, Science, and Technology (MEXT) of Japan [15K148885, 17K08145, and 21K05981] (TK), and by the Japan Agency for Medical Research and Development (AMED) [19fk0108080j0201] (TK). The funders had no role in study design, data collection and analysis, decision to publish, or preparation of the manuscript.

**Competing interests:** The authors have declared that no competing interests exist.

Blockade of IL-17A prevented nuclear translocation of NF-κB p65 in stroma-like cells and epithelial cells in the small intestine but only partially prevented recruitment of immune cells to the small intestine. This study provides an overall picture of the pathogenesis of infection in individual mice at the cytokine and effector levels.

## Author summary

Dengue fever can lead to severe illness in some cases, which can be fatal. The number of severe cases is increasing. There are currently no effective therapeutic drugs or vaccines available for widespread use. Vascular leakage is the physiological basis of the pathology leading to severe disease, and it is widely believed that vascular leakage is directly or indirectly driven by an excessive immune response, that is, a cytokine storm. Although many hypotheses have been proposed, a complete understanding of disease progression has been missing. Therefore, we performed a comprehensive analysis of gene expression in the mouse model and found that the changes in the small intestine are critical for severe disease. In the small intestine, among six types of γδ T cells, IL-17A-producing Vγ4 and Vγ6 T cell receptor (TCR) γδ T cells were selectively expanded following infection. IL-17A played a critical role, through NF-κB activation, in inducing the production of IL-6 and neutrophil collagenase/matrix metalloprotease 8 (not previously considered an inducer of vascular leakage), as well as STAT-3 phosphorylation, acting synergistically with TNF-α to promote vascular leakage. This study both reinforces and challenges some of the previous observations, while highlighting the need for new perspectives in this field.

## Introduction

Severe dengue, known as dengue hemorrhagic fever, is associated with transient vascular leakage from microvessels, leading to life-threatening dengue shock syndrome. Vascular leakage occurs rapidly and generally lasts for 48–72 h [1]. High levels of cytokines, chemokines and activated T cells are detected in patient sera [2]. These observations suggest that severe dengue is caused by excessive activation of host immune cells, a so-called "cytokine storm" [3]. Cytokine storms are considered a major cause of severe disease in viral hemorrhagic fever [4], septic syndrome [5], and acute lung injury during infection with influenza A virus [6] or severe acute respiratory syndrome-coronavirus 2 (SARS-CoV-2) [7]. However, the detailed mechanism underlying a cytokine storm remains unknown.

IL-17A plays a major role in the cytokine storm associated with toxic shock syndrome [8] and acute lung injury induced by influenza infection [9]. IL-17A was originally thought to play a crucial protective role in host defense against fungal and bacterial infections [10]; however, IL-17A is now known to mediate the excessive secretion of proinflammatory cytokines, chemokines, and matrix metalloproteases (MMPs) in synergy with other cytokines such as TNF-α [11–13]. IL-17A also directs neutrophil maturation and differentiation [14]. Higher levels of IL-17A have been detected in dengue patients; however, whether these higher levels [15–18] contribute to severe disease requires further study.

Mouse models of severe dengue have been developed. These models display human symptoms, including vascular leakage [19–22]. In these models, the liver and/or intestine show severe damage, which are common feature of severe dengue in humans [23–25]. Importantly, it has been consistently reported by several research groups (including ours) that blockade of

TNF-α signaling with anti-TNF-α antibodies (Ab) protects mice [19–22], suggesting that the cytokine storm is the cause of severe disease in these mouse models. However, the detailed mechanism underlying the protection afforded by blockade of TNF-α signaling remains unknown.

The pathogenesis of severe dengue can be divided into two steps. The first is the activation of immune cells and the excessive production of cytokines (cytokine-level event). The second is the induction of vascular leakage (effector-level event). For example, the production of TNF-α is a cytokine-level event, and TNF-α can stimulate the expression of many effector genes, such as MMPs (effector-level event). Here, we investigated both cytokine- and effector-level events that drive severe dengue. We performed microarray analysis of the liver and small intestine to study the effect of blockade of TNF-α signaling on infected mice, as these particular organs showed more severe vascular leakage than other organs [22]. We found that IL-17A produced by γδ T cells in the small intestine was another critical factor enabling severe dengue at the cytokine level. Importantly, Vγ4 and Vγ6 T cell receptor (TCR) γδ T cells were selectively expanded in the small intestine of the infected mice. At the effector level, infection induced massive neutrophil infiltration into the lamina propria of the small intestine and produced neutrophil-derived MMP-8. Treatment with anti-TNF-α Ab or anti-IL-17A Ab treatment only partially suppressed the expansion of γδ T cells and neutrophil infiltration in the small intestine, but strongly inhibited the nuclear translocation of NF-kB p65 in small intestinal stroma-like cells.

## Results

### Dengue disease progressed rapidly between days 3 and 4 post-infection with dengue virus (DENV)

We previously reported that infection with type 3 dengue virus (DENV-3) P12/08 causes vascular leakage in the liver and small intestine, leading to death [22]. We first investigated the chronology of the pathological changes occurring after DENV infection to better understand the pathological mechanism of severe illness. Infected IFN-α/β and γ receptor knockout (IFN-α/β/γRKO) mice died on day 4 post-infection (p.i.). Mice were dissected on days 2, 3, and 4 p.i. (Fig 1A). On day 3 p.i., both the liver and small intestine started showing mild gross changes, which became more evident on day 4 p.i. (Fig 1B). The color of the liver faded to white, and the small intestine was swollen. On day 4 p.i., all mice had severe diarrhea. Next, we examined for vascular leakage by injecting Evans blue dye, which revealed that vascular permeability in both the liver and small intestine increased suddenly on day 4 p.i. (Fig 1C). These observations suggest that pathological changes started to occur between days 3 and 4 and progressed rapidly. We also confirmed that anti-TNF-α Ab treatment prevented vascular leakage both in the liver and small intestine on day 4 p.i., although the level of vascular leakage was still high in the liver compared with that of mock-infected mice (Fig 1G and 1H).

### Pathway analysis revealed increased cytokine production in the small intestine, activation of the IL-17 signaling pathway, and induction of MMP-8 expression

To define the gene expression signatures associated with severe disease and the effect of anti-TNF-α Ab treatment, we used a mouse microarray to profile global gene expression in the liver and small intestine in five groups of mice (Groups A to E): DENV-3 P12/08-infected mice with control IgG treatment collected on day 3 (Group A) and 4 p.i. (Group B), DENV-3 P12/08-infected mice with anti-TNF-α Ab treatment collected on day 3 (Group C) and 4 p.i.

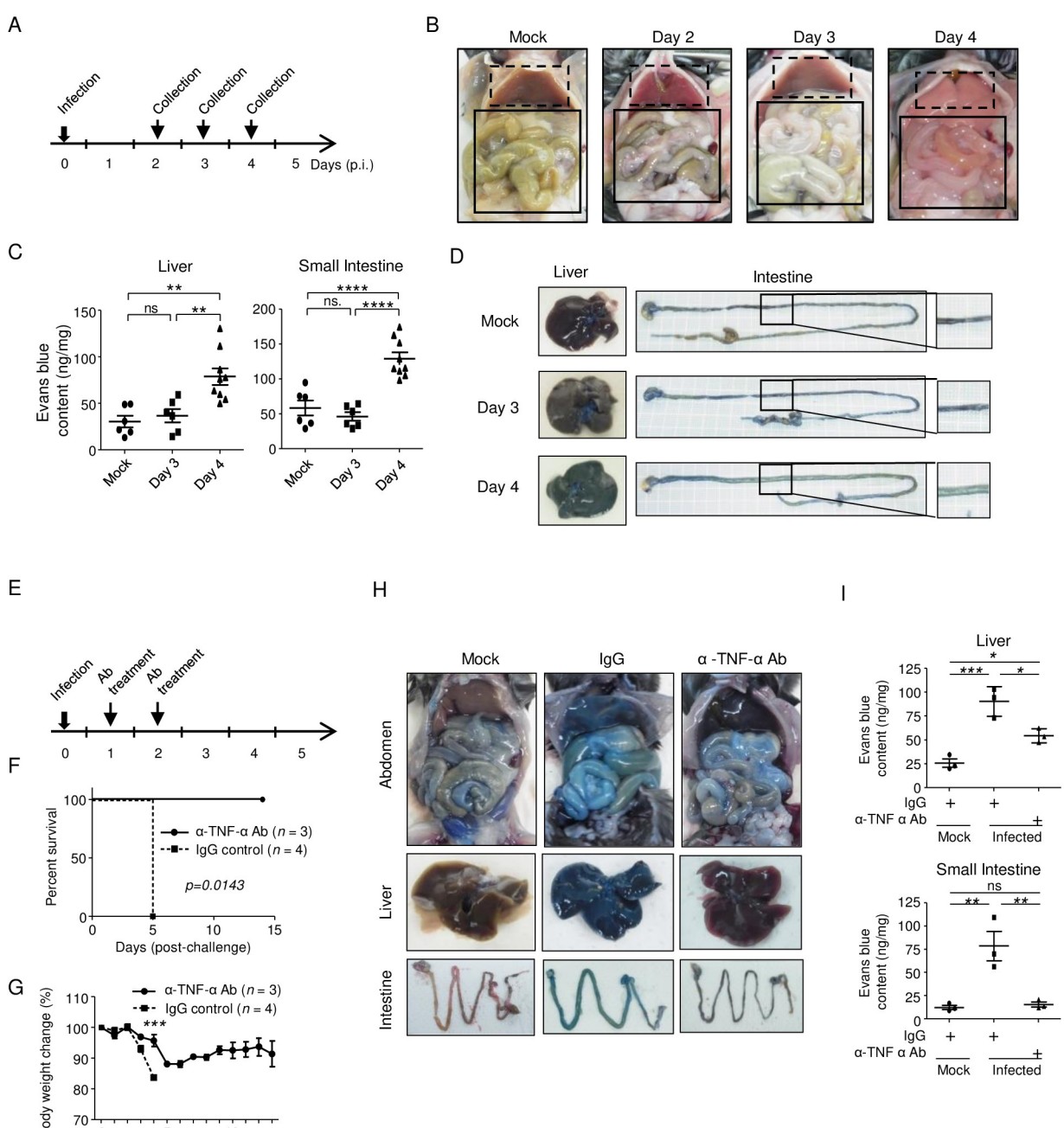

**Fig 1. Disease progression and protection by anti-TNF-α Ab treatment.** (A) IFN-α/β/γR knockout (KO) mice (8–10 weeks old) were infected intraperitoneally with $2 \times 10^6$ focus-forming units (FFU) of DENV-3 P12/08 and sacrificed at Days 2, 3, and 4 post-infection (p.i.). (B) Ventral gross appearance of IFN-α/β/γRKO mice infected with DENV-3 P12/08 or mock infected and the effect of α-TNF-α Ab treatment. The dotted rectangles indicate the liver, and the solid rectangles indicate the small intestine. The experiment was repeated five times with similar results. (C) Groups of mice (n = 6–8/group) were infected intraperitoneally with $2 \times 10^6$ FFU of DENV-3 P12/08 and sacrificed at Day 3 or 4 p.i. IFN-α/β/γRKO mice infected intraperitoneally with mock or $2 \times 10^6$ FFU of DENV-3 P12/08 were treated with IgG or anti-TNF-α Ab, and then injected intravenously with Evans blue 2 h prior to sacrifice at Day 4 p.i. Mice were then perfused extensively with PBS. Evans blue was extracted from the liver and small intestine, and its concentration was measured. The results are expressed as optical density/gram of tissue. Significance was assessed using Tukey's multiple comparison test (n = 6–8/group). *$P < 0.05$, **$P < 0.01$, ***$P < 0.001$, and ****$P < 0.0001$. ns indicates no significant difference. Extravasation of the dye into the liver and small intestine of mock-infected or DENV-3 P12/08-infected mice was measured at Day 3 or 4 p.i. The experiment was repeated twice with similar results. (D) Macroscopic pathology of the liver and small intestine of IFN-α/β/γRKO mice injected with Evans blue. The experiment was repeated five times with similar results. (E) Anti-TNF-α Ab treatment. IFN-α/β/γRKO mice (8–10 weeks old) were infected intraperitoneally with $2 \times 10^6$ FFU of DENV-3 P12/08. Mice were injected with the anti-TNF-α Ab (n = 3) or control normal IgG (n = 4) at Days 1 and 2 p.i., and survival was monitored. (F) Kaplan–Meier survival curves show the percentage of mice surviving at the specified days p.i. Significant differences were evaluated by the log-rank (Mantel–Cox) test. $P = 0.0143$. (G)

Percent body weight change in infected mice from the anti-TNF-α Ab-injected (solid circles) versus IgG-injected (solid squares) groups. The mean body weight of surviving mice is shown as a percentage of the pre-infection weight; bars = ± SEM. Percent body weight change on each day was analyzed using the unpaired t-test. ***$P < 0.001$. (H) Extravasation of Evans blue into the liver and small intestine. The experiment was repeated three times with similar results. (I) Quantification of Evans blue in the liver and small intestine of infected mice with/without anti-TNF-α Ab treatment and mock-infected mice at Day 4 p.i. (n = 3/group). The experiment was repeated twice with similar results. Data were analyzed by one-way ANOVA and the significance of differences was assessed by Tukey's multiple comparison test (n = 3/group). *$P < 0.05$, **$P < 0.01$, ***$P < 0.001$, and ****$P < 0.0001$.

(Group D), and mock-infected mice (Group E) (Fig 2A). Differentially expressed genes (DEGs) were defined as those showing a ≥2-fold change in expression in an unpaired Student's t-test ($p ≤0.01$ for differential expression analysis). We first performed an exploratory study using principal component analysis. In the liver, samples were grouped according to the day of infection (Groups A and C = day 3 p.i.; Groups B and D = day 4 p.i.), although they were still separated according to anti-TNF-α Ab treatment (Fig 2B). In the small intestine, infection was a major cause of variable gene expression (PC1 = 28%). Samples from infected mice collected on day 4 p.i. (Group B) were grouped separately (Fig 2C), suggesting that the induction of significant changes in gene expression on day 4 p.i. was efficiently suppressed by anti-TNF-α Ab treatment. To determine the nature of the DEGs, we performed Gene Set Enrichment Analysis (GSEA) based on Kyoto Encyclopedia of Genes and Genomes (KEGG) gene sets. In the liver, the pathway "cytokine-cytokine receptor interaction" was the highest ranked pathway on day 3 p.i. (Fig 2D) and the number of genes related to this term was larger on day 3 than on day 4. Anti-TNF-α Ab treatment inhibited the changes in the expression of these genes. In the small intestine, the highest ranked pathway was also "cytokine-cytokine receptor interaction" (Fig 2E) and the second highest ranked pathway was "viral protein interactions with cytokine and cytokine receptors". These changes were more pronounced on day 4 than on day 3. These results indicated that cytokines play a major role in disease progression. Although DEGs in mice treated with the anti-TNF-α Ab were still enriched in these pathways, there were fewer in number. Notably, the IL-17 signaling pathway played a critical role in the small intestine (Fig 2E). Similarly, anti-TNF-α Ab treatment reduced the number of DEGs in the IL-17 signaling pathway. Next, we generated a heatmap of the differential expression of genes in the cytokine-cytokine receptor interaction pathway (Fig 2F and 2G). In the livers of infected mice, the expression of most genes was highest on day 3 p.i. (Fig 2F). By contrast, the expression of these genes in the small intestine of infected mice was highest on day 4 p.i., and expression levels decreased after exposure to the anti-TNF-α Ab (Fig 2G). This suggests that infection induces greater cytokine production in the small intestine than in the liver, and that anti-TNF-α Ab treatment is more effective in suppressing gene expression in the small intestine on day 4.

Next, we used Ingenuity Upstream Regulator Analysis to identify regulators responsible for the inflammatory and immune responses, which revealed a high score for a set of upstream cytokine genes in both organs on days 3 and 4 (i.e., *il1β*, *tnf*, *ifn-γ*, *csf2*, *il6*, and *il17a*; *p*-value of overlap < $10^{-15}$) (Fig 3A, 3B and S1 Table). Anti-TNF-α Ab treatment reduced the -log (*p*-value) for most regulators in the liver on day 3 p.i., but not on day 4 p.i. However, in the small intestine, anti-TNF-α Ab treatment reduced the -log (*p*-value) of these cytokines markedly on day 4 p.i. (Fig 3A, 3B and S1 Table). These results indicated that several proinflammatory cytokines were upregulated by infection through the TNF-α signaling pathway.

We also searched for factors responsible for vascular leakage at the effector level. The neutrophil collagenase MMP-8 was ranked in the top 10 genes in the small intestine showing upregulated expression on day 4 p.i. (S2 Table). Therefore, we analyzed peptidases by predicting fold changes in expression using IPA. ClustVis revealed that the expression of MMP-8 in the

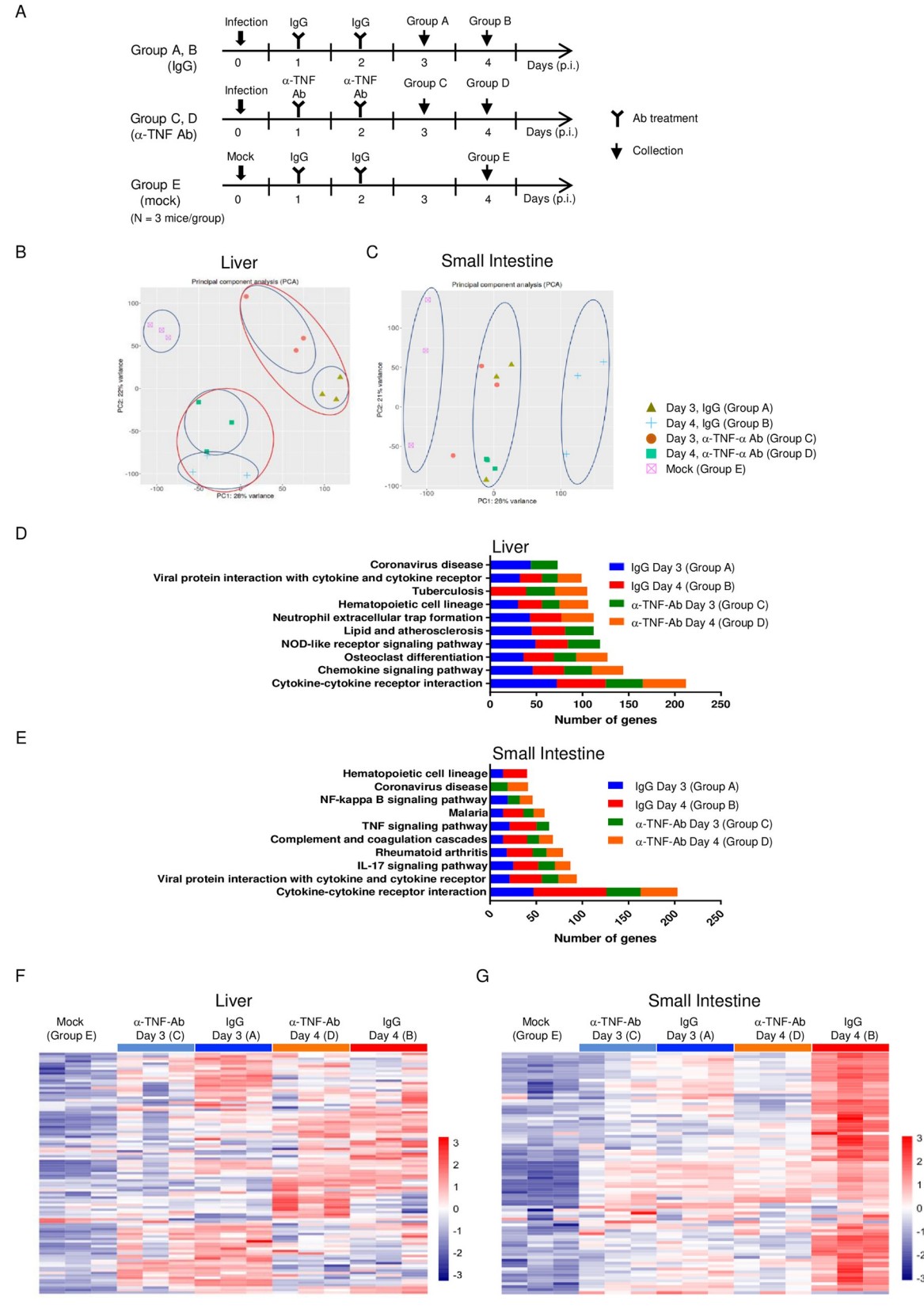

**Fig 2. Microarray analysis in the presence/absence of anti-TNF-α Ab.** IFN-α/β/γR knockout (KO) mice (8–10 weeks old) were infected intraperitoneally with $2 \times 10^6$ focus-forming units (FFU) of DENV-3 P12/08. Mice were injected with an anti-TNF-α Ab (n = 3) or control normal IgG (n = 4) on Days 1 and 2 p.i., and survival was monitored. (A) Five groups (n = 3/group) were subjected to microarray analysis: DENV-3 P12/08-infected mice with normal IgG treatment and samples were collected on Day 3 (Group A) or 4 p.i. (Group B), DENV-3 P12/08-infected mice with anti-TNF-α Ab treatment and samples were collected on Day 3 (Group C) or 4 p.i. (Group D), and mock-infected mice (Group E). (B and C) Principal component analysis (PCA) of microarray samples of the liver (B) and small intestine (C). (D and E) Number of upregulated genes in the liver (D) or small intestine in enriched KEGG pathways in the following four groups: mice were infected and treated with IgG, and samples were collected on Day 3 (Group A); mice were infected and treated with IgG, and samples were collected on Day 4 (Group B); mice were infected and treated with α-TNF-α Ab, and samples were collected on Day 3 (Group C); and mice were infected and treated with α-TNF-α Ab, and samples were collected on Day 4 (Group D). (F and G) Heatmap of cytokine gene expression in the liver (F) and small intestine (G).

liver and small intestine was highest on day 4 p.i, and that its expression was significantly suppressed by anti-TNF-α Ab treatment (Fig 3C and 3D). In addition, expression of MMP-7 or MMP-3 in the liver or small intestine, respectively, increased upon infection, and this was suppressed by anti-TNF-α Ab treatment. Thus, these MMPs are candidate effector molecules that trigger vascular leakage. By quantitative RT-PCR (qRT-PCR), we confirmed that IL-6, IL-1β, and MMP-8 mRNAs were commonly upregulated in the liver and small intestine after infection and were all suppressed by anti-TNF-α Ab treatment (S1A and S1L Fig). Notably, the small intestine showed a particularly high level of IL-6. We also confirmed that anti-TNF-α Ab treatment had no effect on virus production in the liver, small intestine, or spleen (S1M and S1O Fig).

## DENV infection resulted in increased blood levels of most cytokines and chemokines

Circulating levels of cytokines, chemokines, MMPs, and related factors in sera collected on day 4 p.i. were measured using a Multiplex Luminex Assay. The levels of TNF-α, IL-6, and IL-17A increased significantly (S2A and S2C Fig); but were significantly reduced by anti-TNF-α Ab treatment. Similarly, infection with DENV-3 P12/08 increased MMP-8 levels, which were suppressed by anti-TNF-α Ab treatment (S2D Fig). There was no increase in the level of MMP-7 or MMP-9 after infection (S2E and S2F Fig). We also examined the levels of other cytokines, chemokines, and factors related to the stability of the endothelium. The levels of IL-12p70, MCP-1, IL-10, IL-1β, IFN-γ, IL-2, M-CSF, VEGF, ICAM-1, and P-selectin increased after infection; however, the expression of all except MCP-1, IFN-γ, and P-selectin was strongly suppressed by anti-TNF-α Ab treatment (S2G and S2Q Fig). Expression of syndecan-1 increased slightly after infection (S2R Fig). This suggests that the levels of most inflammatory cytokines, anti-inflammatory cytokines, chemokines, and soluble adhesion molecules increase after infection.

## Blockade of IL-17A signaling, but not IL-6 signaling, partially protected mice from lethal infection with DENV

The above results indicated the strong involvement of IL-6 or IL-17A, in addition to TNF-α, in severe dengue at the cytokine level. The direct inhibition of the IL-6 molecule by MP5-20F3 antibody showed only a weak protective effect [22]. Therefore, to block IL-6 signaling, we treated infected mice with an anti-IL-6 receptor (R) Ab, MR16-1; however, this had no protective effect (Fig 4A and 4B). We questioned whether this was because the antibody did not effectively suppress IL-6 signaling because of the extremely high levels of IL-6 in serum (S2B Fig). Serum albumin A levels, which reflect the activity of IL-6 [26], were particularly high, even after anti-IL-6R Ab treatment (Fig 4C), suggesting insufficient blockade of IL-6 signaling.

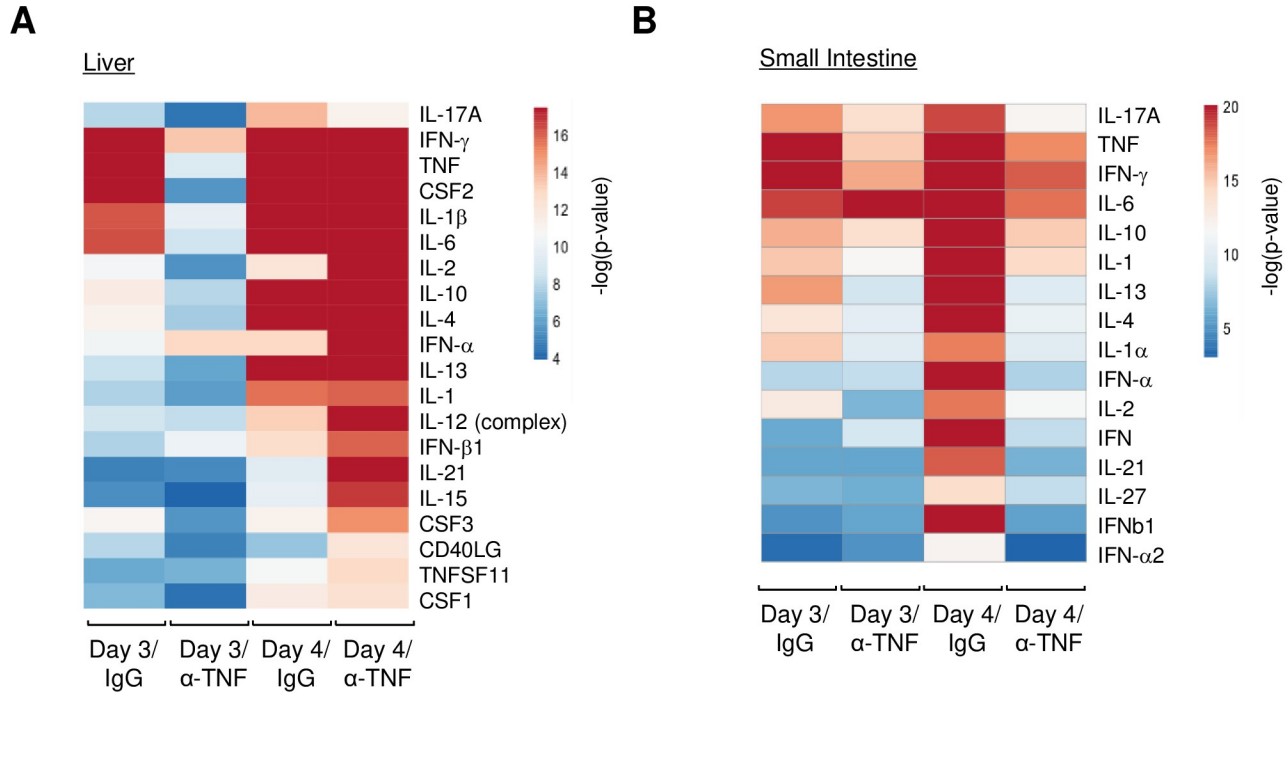

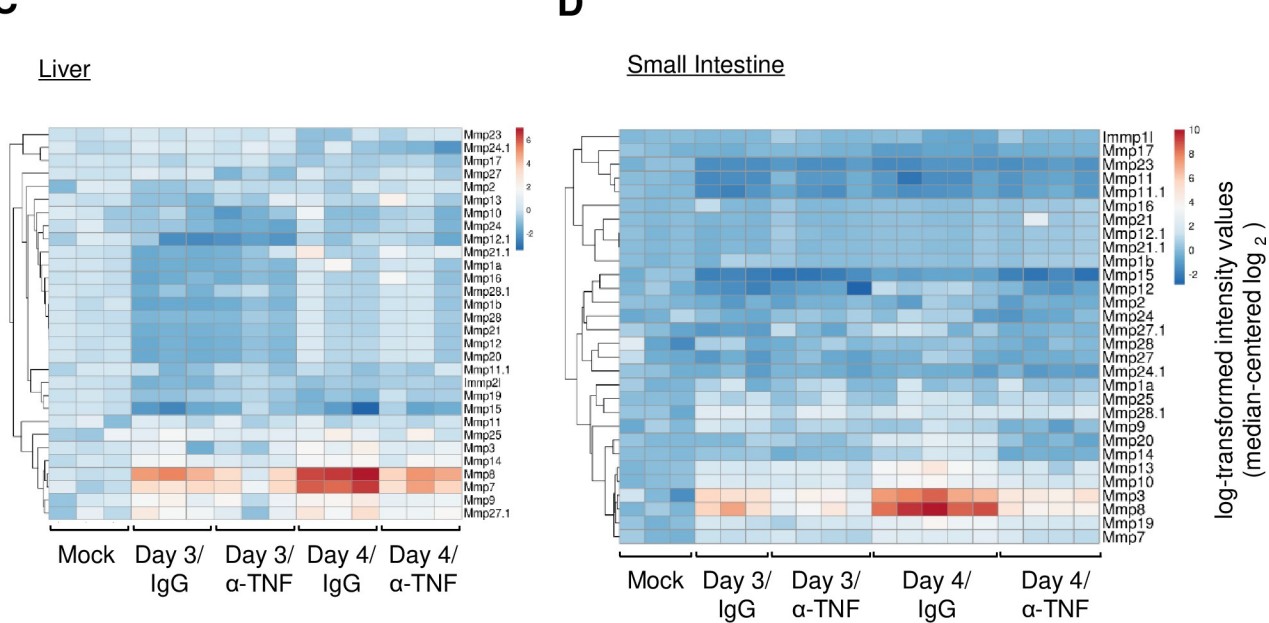

**Fig 3. IPA-predicted upstream regulators of gene expression.** (A and B) IFN-α/β/γR knockout (KO) mice (8–10 weeks old; n = 3) were injected with rat normal IgG or an α-TNF-α Ab on Days 1 and 2 post-infection (p.i.). The liver (A) and small intestine (B) were collected on Day 3 or 4 p.i. Heatmap of transcription factors predicted to be activated and inhibited upstream of genes that showed altered expression in this study. Upstream regulators were categorized as cytokines by IPA. In both heatmaps, colors indicate the −log P value. A default cut-off log P value < 12 was used in these analyses. The complete data with P values are shown in S1 Table. (C and D) Heatmap of MMP genes in the liver (C) and small intestine (D). Heatmaps are based on the expression levels of each MMP gene (in ClustVis). The heatmap shows relative expression of MMP genes normalized to that in mock-infected mice. Fold changes in activation of MMP genes in individual mice are indicated by different colors. The color scale depicts changes in expression.

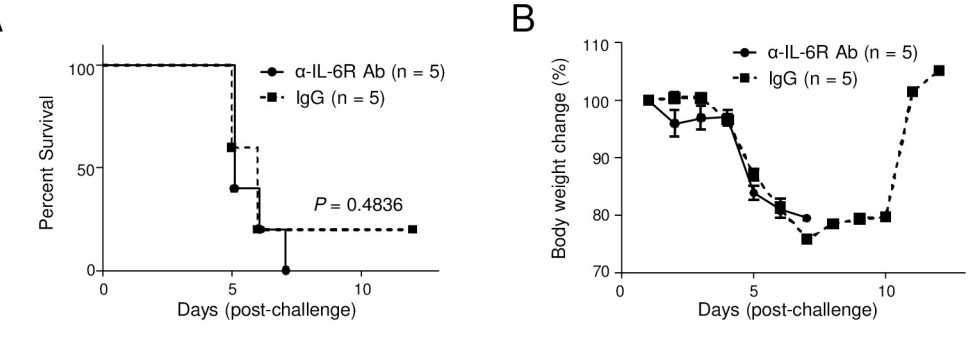

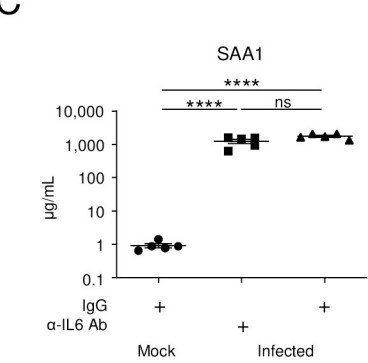

**Fig 4. Effect of anti-IL-6R Ab on mice.** (A) Mouse survival. Eight-week-old IFN-α/β/γR knockout (KO) mice were infected with $2 \times 10^6$ of DENV-3 P12/08 or mock (n = 5). A total of 8 mg of anti-IL-6R Ab (n = 5) or isotype control IgG (n = 5) was inoculated intraperitoneally at Day 1 p.i., and the rate of survival was observed until Day 12 p.i. Kaplan–Meier survival curves show the percentage of mice surviving at the specified days p.i. Significant differences between individual groups were evaluated by the log-rank (Mantel–Cox) test. P = 0.4836. (B) Relative body weight. Relative body weight was calculated and is shown as the mean and SEM. (C) Serum albumin A (SAA) concentrations in treated mice. Serum samples were collected for analysis by ELISA at at Day 4 p.i. from infected IFN-α/β/γRKO mice with IL-6R Ab or IgG and from mock-infected IFN-α/β/γRKO mice. Results are expressed as the mean ± SEM. Significance was assessed using Tukey's multiple comparison test.

Next, we examined blockade of IL-17A signaling. Treatment with an anti-IL-17A Ab protected 50% of mice from lethal infection (Fig 5A), although their body weight decreased (Fig 5B). In addition, anti-IL-17A Ab treatment reduced vascular leakage significantly (Fig 5C and 5D). We also examined the levels of inflammatory cytokines in serum on day 4 p.i. We found that α-IL-17A Ab treatment reduced the levels of TNF-α and IL-6 to those in mice treated with the anti-TNF-α Ab (Fig 5E and 5F). Unexpectedly, anti-IL-17A Ab treatment increased IL-17A levels (Fig 5G). In addition, the α-IL-17A Ab completely suppressed IL-1β (Fig 5H) and IL-12p70 (Fig 5J), but not IL-10 or IFN-γ (Fig 5I and 5K). Notably, the α-IL-17A Ab also suppressed MMP-8 (Fig 5L), but not MMP-3 (Fig 5M), suggesting that MMP-3 plays less of a role in severe dengue. The α-IL-17A Ab did not affect viral RNA levels in the liver, small intestine, or spleen (S3A and S3C Fig).

## IL-17A was produced by γδ T cells in the small intestine and Vγ4 and Vγ6TCR γδ T cells were selectively expanded following DENV infection

Although Th17 cells are known to produce IL-17A in the small intestine [27], we suspected that γδ T, CD8+ T, or ILC3 cells, but not Th17 cells, were the likely producers of IL-17A in our model because of the rapidity of cytokine production. To identify IL-17A-producing cells, we isolated cells from the small intestine and examined a plot of side scatter (SSC) versus CD45

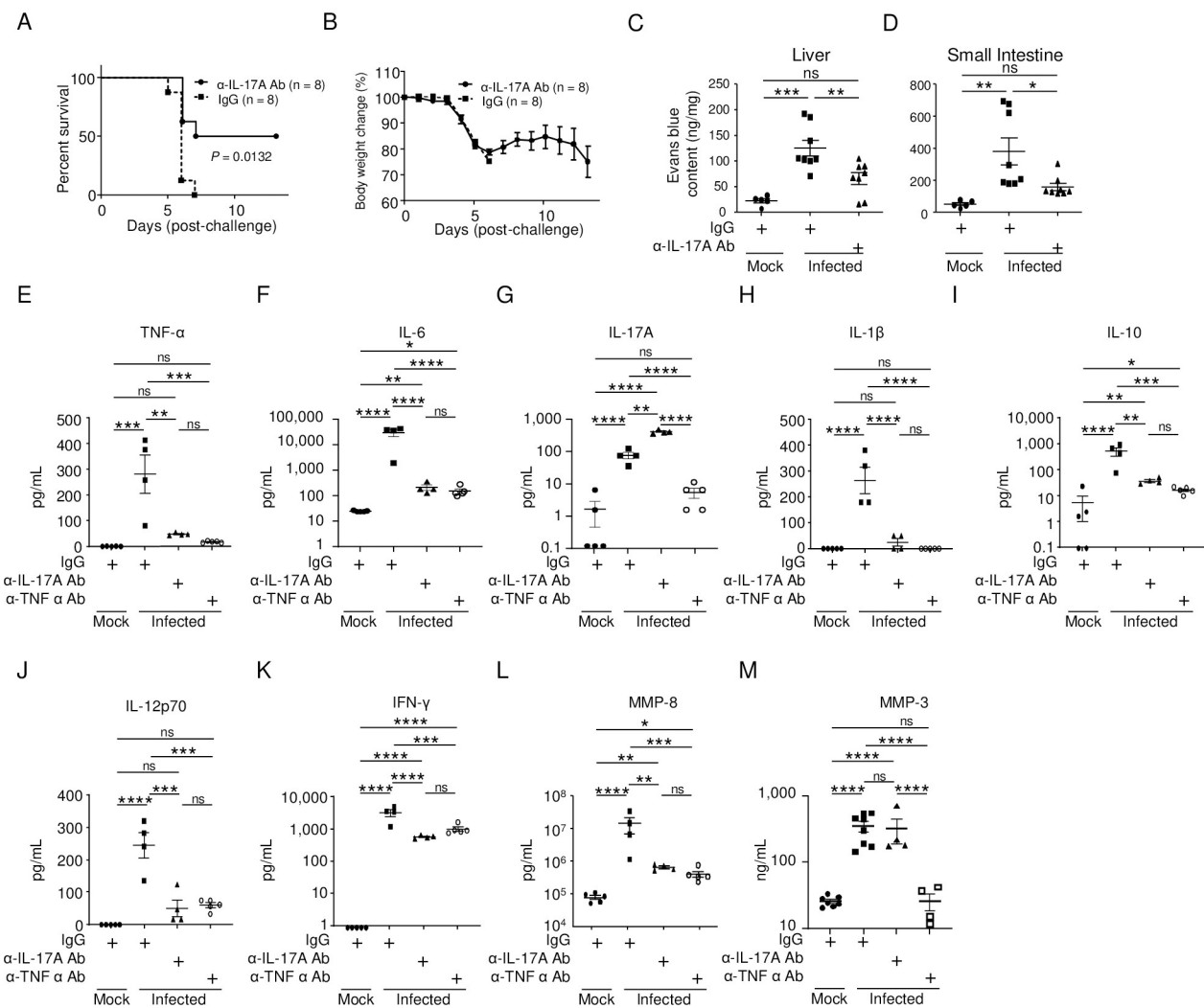

**Fig 5. Administration of anti-IL-17A Ab in mice.** (A) IFN-α/β/γR knockout (KO) mice (8–10 weeks old) were infected with $2 \times 10^6$ focus-forming units (FFU) of DENV-3 P12/08. Anti-IL-17A Ab (n = 8) or isotype control IgG (n = 8) was injected intraperitoneally on Days 1 and 2 p.i., and the mice were observed up until Day 13 p.i. Kaplan–Meier survival curves show the percentage of mice surviving on the specified days p.i. Significant differences between individual groups were evaluated by the log-rank (Mantel–Cox) test. $P = 0.0132$. (B) Percent body weight change was calculated and is shown as the mean ± SEM. The experiment was repeated three times with similar results. (C) Mock-infected mice (closed circles, n = 5), mice injected with IgG (closed squares, n = 8), and mice injected with an α-TNF-α Ab (closed triangles, n = 8) were injected intravenously with Evans blue dye on Day 4 p.i. The mice were then sacrificed under anesthesia and perfused extensively with PBS. (C and D) Extravasation of the Evans blue dye into the peritoneal cavity, liver (C), and small intestine (D) of mock-infected or DENV-3 P12/08-infected mice was measured after recovering the dye. (E–M) Serum host factor concentrations. IFN-α/β/γRKO mice (8–10 weeks old; n = 4–5) were infected intraperitoneally with $2 \times 10^6$ FFU of DENV-3 P12/08, followed by intraperitoneal injection of anti-IL-17A Ab or isotype control IgG on Days 1 and 2 p.i. The mice were sacrificed under anesthesia on Day 4 p.i. and sera were collected. Serum concentrations of TNF-α (E), IL-6 (F), IL-17A (G), IL-1β (H), IL-10 (I), IL-12p70 (J), IFN-γ (K), MMP-8 (L), and MMP-3 (M) were measured using a Mouse Magnetic Luminex Assay kit. (N–U) Total RNA was extracted from the liver (n = 4–5) and small intestine (n = 4–5) and subjected to quantitative RT-PCR. The expression of each mRNA was calculated relative to that in mock-infected mice. Levels of mRNA expression of TNF-α (N and O), IL-6 (P and Q), IL-17A (R and S), and MMP-8 (T and U) in the liver (N, P, R, and T) and small intestine (O, Q, S, and U) of mock- (8–10 weeks old) and DENV-3 P12/08 ($2 \times 10^6$ focus-forming units [FFU])-infected mice. Data were analyzed by one-way ANOVA and the significance of differences was assessed by Tukey's multiple comparison test. $^*P < 0.05$, $^{**}P < 0.01$, $^{***}P < 0.001$, and $^{****}P < 0.0001$.

(Fig 6A). This plot showed an increase in the CD45$^+$ cell population, which includes any type of white blood cell, after infection (Fig 6A). Gated CD45$^+$ leukocytes were further analyzed by measuring CD3 versus γδTCR signals. Three major cell populations were observed in infected mice: CD3$^+$/γδTCR$^+$ cells, CD3$^+$/γδTCR$^-$ cells, and CD3$^-$/γδTCR$^-$ cells (Fig 6B). Among these,

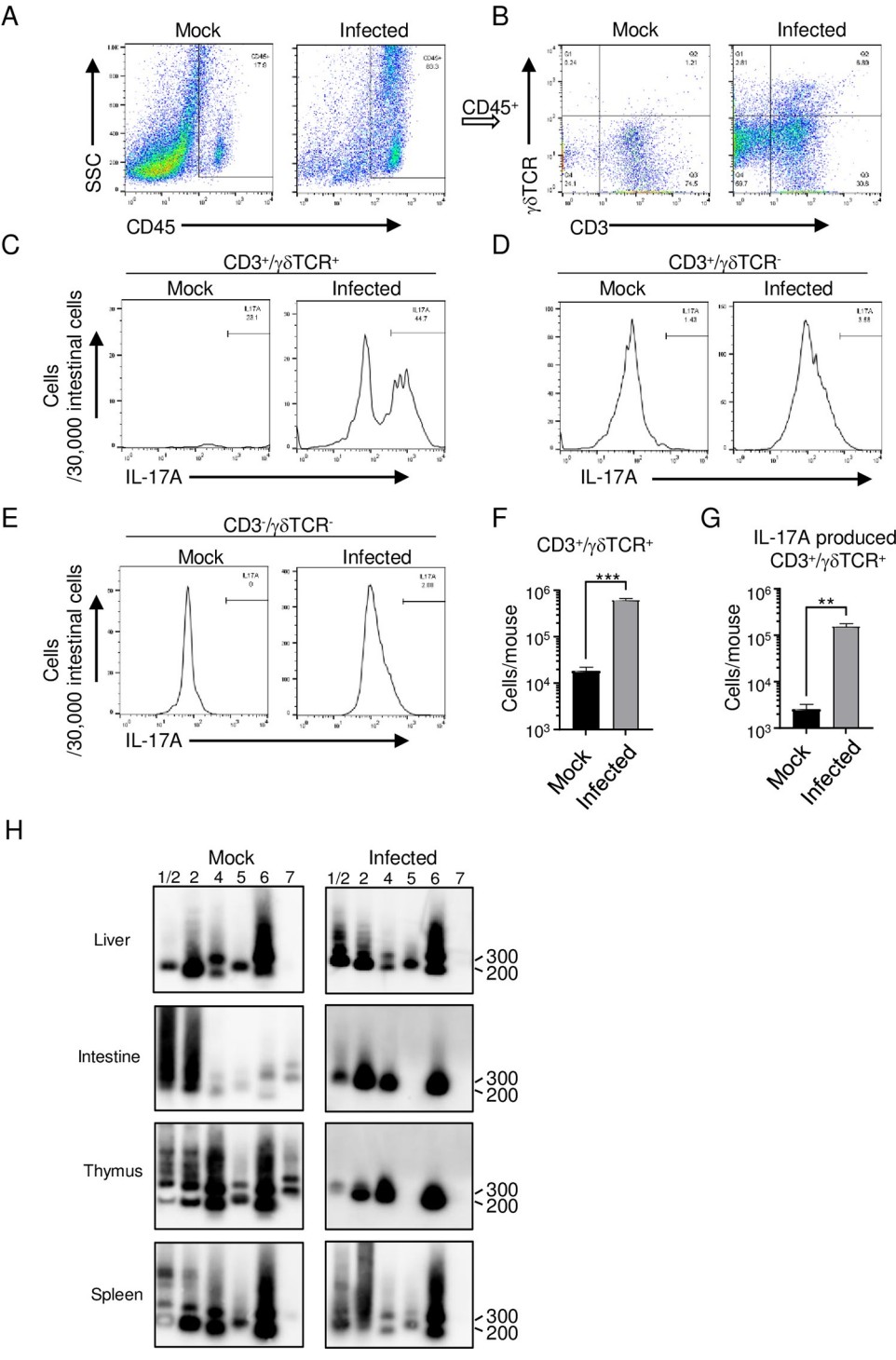

**Fig 6. Expansion of IL-17A-producing Vγ4 and Vγ6 γδ T cells in the small intestine.** IFN-α/β/γR knockout (KO) mice (8–10 weeks old; n = 3) were infected with $2 \times 10^6$ focus-forming units (FFU) of DENV-3 P12/08 or mock, and sacrificed on Day 4 post-infection. Representative flow cytometry plots of gated intestinal cells are shown. Discrimination of CD45+ cells from mock-infected mice (left panel) or DENV-3 P12/08-infected mice (right panel) is shown. (A) CD45+ cells were characterized by the presence of CD3 and the γδTCR receptor. (B) IL-17A-expressing γδ T cells (CD45+CD3+γδT+), CD45+CD3+γδ T− cells (C), and CD45+CD3−γδ T cells (D). (E) The numbers in each quadrant represent the percentage of all gated events. (F) Quantification of the total number of γδ T cells and (G) total number of IL-17A-producing γδ T cells. Experiments were repeated two times in triplicate. (H) Vγ use by γδ T cells isolated from the liver, small intestine, thymus, and spleen on Day 4 post-infection of IFN-α/β/γRKO mice with mock

or DENV-3 P12/08. Total RNA extracted from γδ T cells isolated from α/β/γRKO mice infected with DENV-3 P12/08 was reverse transcribed into cDNA and amplified by PCR using primers specific for Cγ and various Vγ segments. A Southern blot of the γ PCR products was hybridized with an oligonucleotide probe specific for Cγ. Results are representative of three independent experiments.

only CD3$^+$/γδTCR$^+$ cells, considered as γδ T cells, expressed IL-17A (Fig 6C); the CD3$^+$/γδTCR$^-$ cells (including CD8$^+$ T cells) (Fig 6D) and CD3$^-$/γδTCR$^-$ cells (including ILC3 cells) did not express IL-17A (Fig 6E). Of note, the negative IL-17A signal peak in mock-infected mice was small because of the small number of γδ T cells in the small intestine of mock-infected mice, as shown in Fig 6F. In addition, the number of IL-17A-producing γδ T cells increased after infection (Fig 6G), and these cells were the major IL-17A producers in our model.

The Vγ gene of the mouse γδ TCR is encoded by six genes (Vγ1, Vγ2, Vγ4, Vγ5, Vγ6, and Vγ7) [28] that determine the type of cytokine produced [29], and there is a strong correlation between γδ gene usage and tissue localization [30]. Therefore, to understand this cell population, γδ T cells were further subdivided into subsets based on their expression of certain TCR Vγ-chains. γδ T cells were isolated from the liver, small intestine, thymus, and spleen, and the Vγ genes were amplified by RT-PCR and analyzed by Southern blotting. Small intestinal γδ T cells from mock-infected mice mainly expressed the Vγ1/2 and 2 genes, whereas those from infected mice expressed Vγ1/2, 2, 4, and 6 genes (Fig 6H). Thymic γδ T cells from mock-infected mice expressed all Vγ genes, whereas those from infected mice expressed Vγ1/2, 2, 4, and 6. The expression pattern of Vγ genes in the thymus of infected mice was similar to that of intestinal γδ T cells. Infection did not change the population of γδ T cells in the liver and spleen. Collectively, these data suggest that infection induced the selective expansion of Vγ4 and Vγ6TCR γδ T cells in the small intestine.

## Doxycycline prolonged mouse survival and prevented vascular leakage without changing MMP-8 mRNA levels

The above results indicated the importance of MMP-8 at the effector level. Therefore, we next investigated whether inhibiting MMPs would protect mice against lethal infection. Infected mice were injected intraperitoneally once a day with doxycycline, a broad-spectrum MMP inhibitor (Fig 7A). All infected mice treated with phosphate-buffered saline, as a control, died on day 4 p.i., whereas 75% of doxycycline-treated mice survived for an additional 2 days (Fig 7B). Although infected mice treated with doxycycline showed a loss of body weight (Fig 7C), no symptoms such as a hunched posture with ruffled fur or diarrhea were observed on day 4 p.i. Furthermore, doxycycline treatment suppressed vascular leakage on day 4 p.i. (Fig 7D and 7E). We questioned whether the protective properties of doxycycline were a result of its antibacterial effects. It has been reported that interactions between the microbiota and γδ T cells can influence disease pathology, and bacterial cues may promote the expansion of IL-17A-producing γδ T cells [31]. However, administration of an antibiotic cocktail 2 weeks prior to challenge did not improve survival (S4 Fig). Furthermore, doxycycline did not alter the expression of MMP-8 mRNA (Fig 7F and 7G) or IL-6 (Fig 7H and 7I) in either organ, suggesting that the protective effect was due to the inhibition of downstream effector events in these pathways. Doxycycline has also been reported to inhibit DENV protease and replication in cultured cells [32]. However, in our study, doxycycline treatment did not alter virus production in the liver, small intestine, or serum (Fig 7J and 7L).

## Infection induced the infiltration of neutrophils and monocytes/macrophages into the small intestine

MMP-8 is primarily produced by neutrophils [33]. Therefore, we examined infiltrating cells in the small intestine by flow cytometry analysis (Fig 8). The number of CD11b$^+$Ly6G$^+$

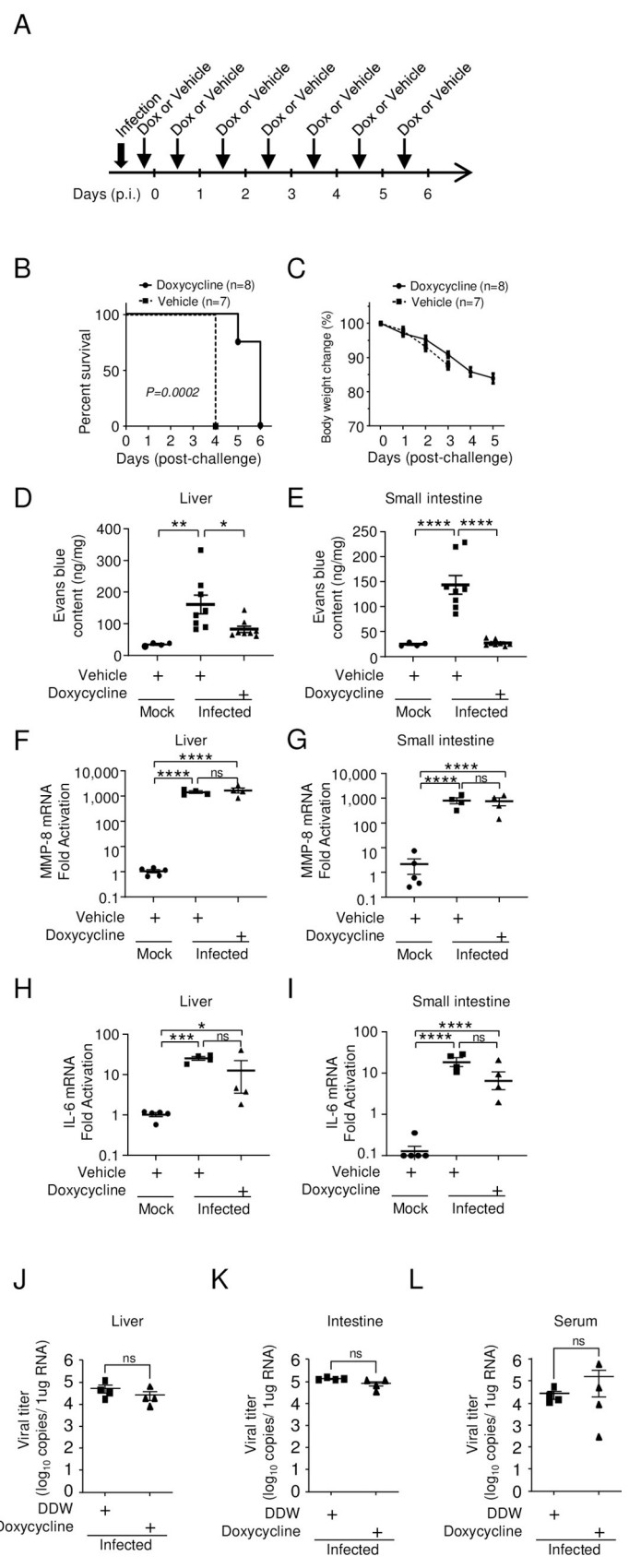

**Fig 7. Prolongation of survival by doxycycline.** (A) IFN-α/β/γR knockout (KO) mice (8–10 weeks old; n = 7–8) were infected with $2.0 \times 10^6$ focus-forming units (FFU) of DENV-3 P12/08 and injected daily intraperitoneally (i.p.) with 2.25 mg of doxycycline. (B) Survival was monitored until Day 6 post-infection (p.i.). Kaplan–Meier survival curves show the percentage of mice surviving at the specified days p.i. Significant differences between two groups were evaluated by the log-rank (Mantel–Cox) test. $P = 0.0002$. The experiment was repeated three times with similar results. (C) Percent body weight change was calculated and is shown as the mean ± SEM. (D and E) Mock-infected mice (closed circles, n = 4), mice treated with distilled deionized water (closed squares, n = 8), and mice treated with doxycycline (closed triangles, n = 8) were injected intravenously with Evans blue dye on Day 4 p.i., sacrificed under anesthesia, and perfused extensively with PBS. Evans blue in the liver (D) and small intestine (E) of mock-infected mice (n = 4) and infected mice treated with/without doxycycline (n = 8/group) was quantified. Data were analyzed by one-way ANOVA and the significance of differences was assessed by Tukey's multiple comparison test. (F–I) On Day 4 p.i., total RNA was isolated from the liver (n = 4–5/group) (F) and small intestine (n = 4–5/group) (G) of mock-infected or virus-infected IFN-α/β/γRKO mice treated with/without doxycycline and subjected to quantitative RT-PCR to detect MMP-8 (F and G) and IL-6 (H and I) mRNA expression. (J–L) Viral titers in the liver (J), small intestine (K), and spleen (L) were measured by quantitative RT-PCR. The results are expressed as the mean ± SEM. Each symbol represents an individual mouse. Data were analyzed by one-way ANOVA and significance was assessed by Tukey's multiple comparison test. *$P < 0.05$, **$P < 0.01$, ***$P < 0.001$, and ****$P < 0.0001$. The experiment was repeated three times with similar results.

neutrophils increased significantly after infection (44.8-fold). In addition, the numbers of CD11b+Ly6G$^{int}$LyF4/80+Ly6C$^{high}$ cells, which are activated by inflammation [34] and thought to be detrimental, and CD11b+Ly6G$^{int}$F4/80+Ly6C$^{low}$ cells, which are thought to be protective [35] monocyte/macrophage subsets, both increased (124.6-fold and 23.3-fold, respectively).

## Histopathological changes in infected mice treated with an anti-TNF-α Ab, an α-IL-17A Ab, or doxycycline

Histopathological analysis of tissue sections stained with hematoxylin–eosin (H&E) revealed changes in the liver and small intestine on day 4 p.i. following treatment. Significant hydropic degeneration with numerous intracellular vacuoles in hepatocytes was observed in infected mice treated with normal IgG (Fig 9A). There was no clear restoration of hepatocytes in IFN-α/β/γRKO mice treated with the anti-TNF-α Ab or α-IL-17A Ab (Fig 9A). Interestingly, almost no damage was observed in the livers of mice treated with doxycycline. By contrast, severe subepithelial edema and infiltration of inflammatory cells in the lamina propria were observed in the small intestines of infected mice treated with the IgG control (Fig 9B). Treatment with anti-TNF-α Ab, anti-IL-17A Ab, or doxycycline partially palliated the intestinal damage, although minor edematous changes were still present. Infiltration of inflammatory cells was partially reduced by each of the three treatments. Staining with α-Ly6b Ab, a marker for monocytes, neutrophils, and dendritic cells [36], showed similar results (Fig 9C). To investigate the structure of the tight junctions, we stained samples for the ubiquitously expressed protein claudin-3 [37]. Claudin-3 expression was strong at cell–cell junctions between small intestinal epithelial cells (Fig 9D); however, the claudin-3 signal decreased and junctional staining became unclear after infection, suggesting disruption of the tight junctions. Treatment with the anti-TNF-α Ab or doxycycline effectively prevented the reduction in claudin-3 expression, while anti-IL-17A Ab treatment achieved only a partial effect, possibly reflecting the weaker protection by α-IL17A Ab on day 4 p.i. (Fig 5A). Although the disappearance of claudin-3 may not exactly reflect disruption of the tight junctions between endothelial cells in microvessels, it suggests breakdown of the physiological structure of cell–cell junctions, which may lead to vascular leakage. We then investigated infiltrated cells by flow cytometry analysis. Anti-TNF-α Ab treatment partly suppressed infiltration of γδ T cells, neutrophils, and CD11b+Ly6G$^{int}$F4/80+Ly6C$^{high}$ cells (6.4-, 4.6-, and 7.3-fold reductions, respectively), but not CD11b+Ly6G$^{int}$F4/80+Ly6C$^{low}$ cells (Fig 9E and 9H), whereas anti-IL-17A Ab or doxycycline

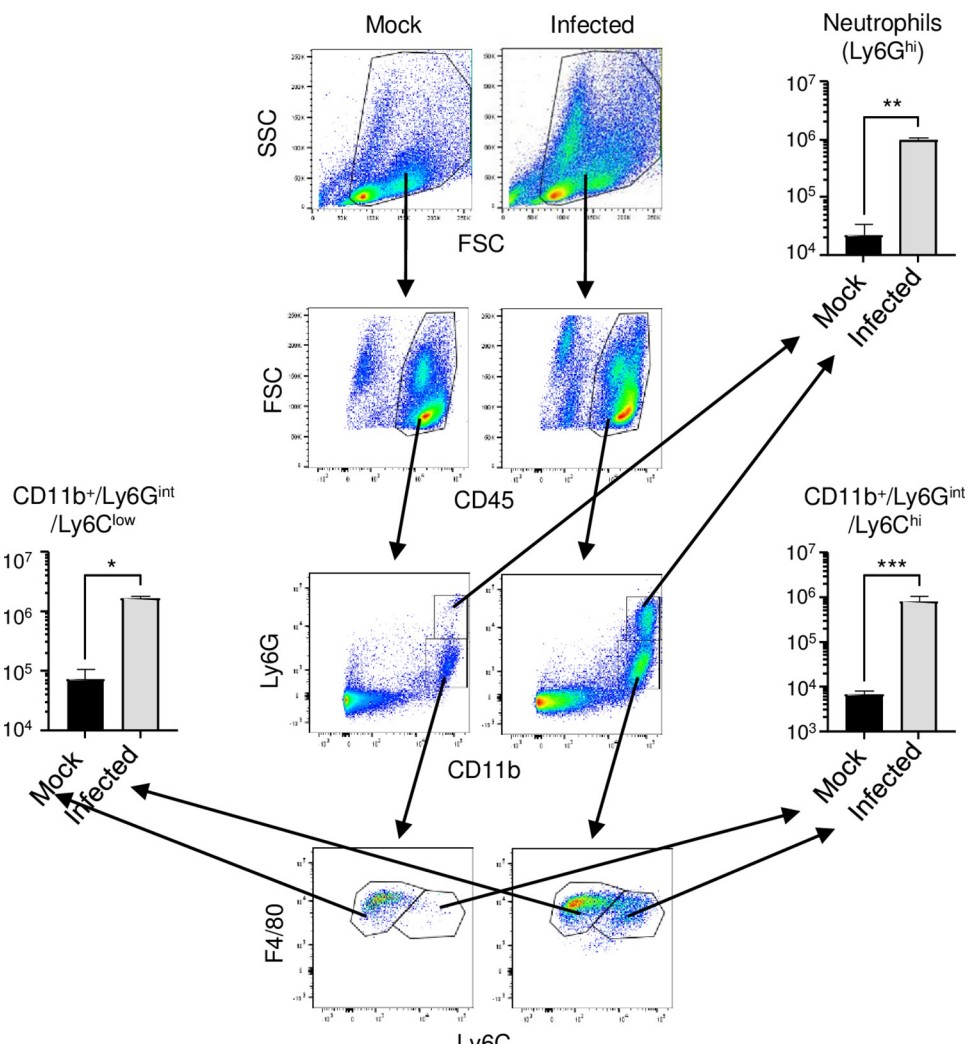

**Fig 8. Analysis of infiltrated neutrophils and monocytes-macrophages in the small intestine of infected mice.**
Representative flow cytometry plots of gated intestinal cells. IFN-α/β/γR knockout (KO) mice (8–10 weeks old; n = 3/ group) were intraperitoneally infected with $2 \times 10^6$ focus-forming units (FFU) of DENV-3 P12/08 or mock, and sacrificed under anesthesia at Day 4 post-infection (p.i.), and the small intestine was collected. The SSC-FSC profile was used to distinguish leukocyte populations from other cell populations. Duplets were removed by singlet gating and debris was removed from the analysis. Leukocyte populations were further gated by FSC-CD45 analysis. CD45+ leukocyte populations were gated by Ly6G-CD11b analysis and neutrophils were classified as CD11b+Ly6Ghi. CD11b+CLy6Gint populations (monocytes/macrophages) were further analyzed and classified into two populations, CD11b+Ly6GintLyF4/80+Ly6Chi and CD11b+Ly6GintF4/80+Ly6Clow populations. The total numbers of CD11b+Ly6Ghi neutrophils, CD11b+Ly6GintLyF4/80+Ly6Chi monocytes, and CD11b+Ly6GintF4/80+Ly6Clow monocytes were quantified. Each symbol represents an individual mouse. Cell numbers were analyzed by one-way ANOVA. Significance was assessed by Tukey's multiple comparison test. *P < 0.05, **P < 0.01, ***P < 0.001 and ****P < 0.0001. The results are expressed as the mean ± SEM. The experiment was repeated three times with similar results.

treatment did not significantly reduce infiltration by these immune cells. However, it remained to be determined how these treatments protected mice.

Cooperation between TNF-α and IL-17A is known to promote the production of proin-flammatory cytokines through the activation of NF-κB [38]. NF-κB is a strong transcriptional activator for IL-6, and translocation of NF-κB p65 protein to the nucleus is required for its activity [39, 40]. We observed strong NF-κB p65 signals in stroma-like cells, including

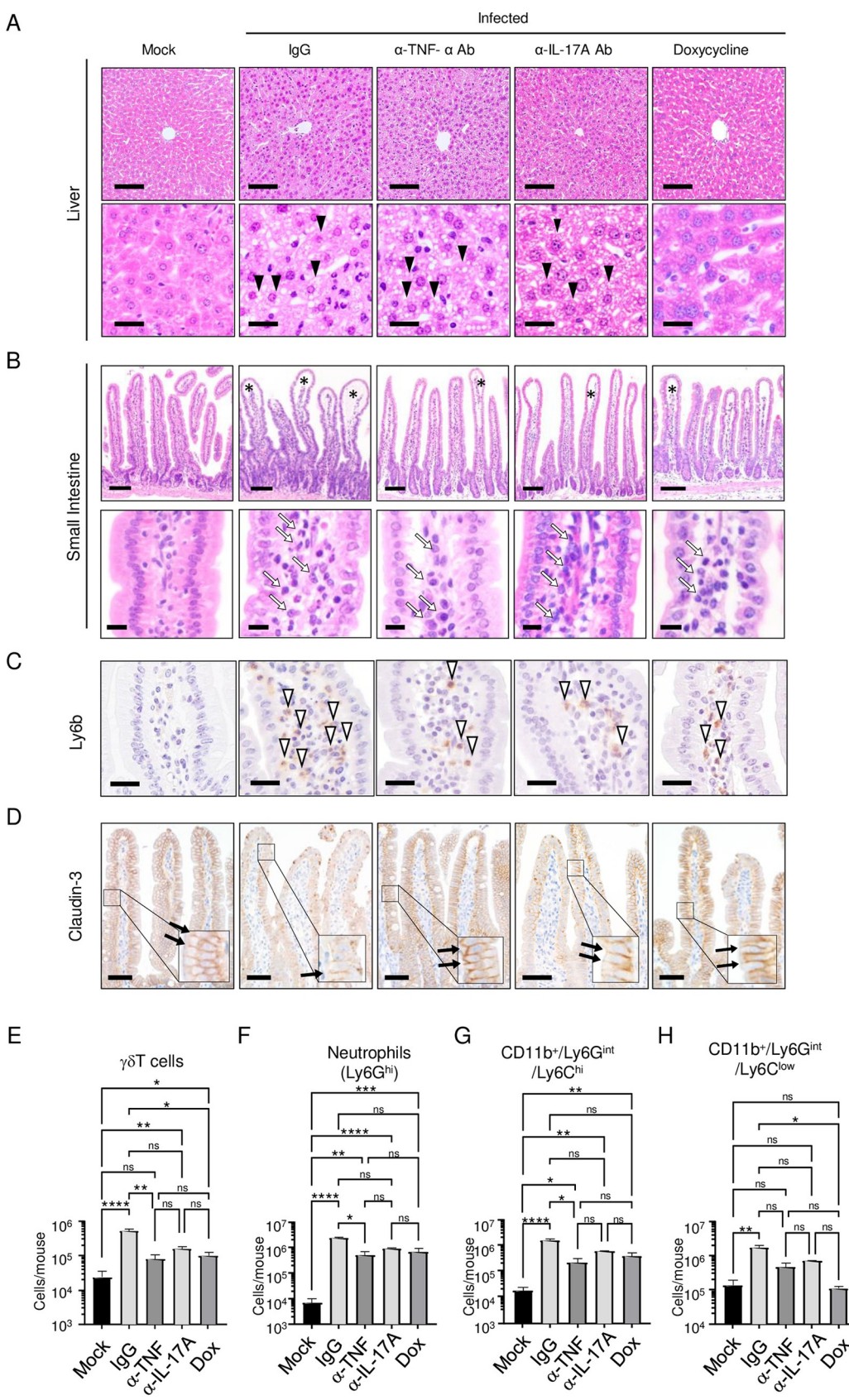

**Fig 9. Histopathological examination of tissues from IFN-α/β/γR knockout (KO) mice.** (A) IFN-α/β/γRKO mice (8–10 weeks old) infected with $2 \times 10^6$ focus-forming units (FFU) of DV3P12/08 or mock-infected mice with PBS were treated with IgG, an α-TNF-α Ab, an α-IL-17A Ab, or doxycycline, and were then euthanized on Day 4 post-infection. Sections of liver (A) and small intestine (B) were prepared, stained with hematoxylin and eosin, and observed under low (× 40) and high (× 400) magnification. Black arrowheads indicate hepatocytes with hydropic degeneration containing numerous vacuoles. Asterisks indicate subepithelial edema in the small intestine of α/β/γRKO-infected mice. Black arrows indicate infiltrated cells in the lamina propria. Bars indicate 100 μm (upper panels) and 25 μm (lower panels) in the liver, and 100 μm (upper panels) in the small intestine (A and B). Intestinal sections were subjected to immunostaining with an anti-Ly6b Ab (C) or an anti-claudin-3 Ab (D). Open arrowheads indicate Ly6b-stained cells and black arrows indicate claudin-3-stained tight junctions. Bars = 25 μm (Ly6b) and 50 μm (claudin-3) (C–D). Images are representative of at least three sections from each tissue. Quantification of the total number of γδ T cells (E), neutrophils (F), CD11b⁺Ly6GintLyF4/80⁺Ly6Chigh monocytes (G), and CD11b⁺Ly6GintF4/80⁺Ly6Clow cells (H) in the small intestine of mock or infected mice treated with IgG, anti-TNF-α Ab, anti-IL-17A Ab, or doxycycline (n = 4–5/group). Data were analyzed by one-way ANOVA and the significance of differences was assessed by Tukey's multiple comparison test. *$P < 0.05$, **$P < 0.01$, ***$P < 0.001$, and ****$P < 0.0001$. Results are expressed as the mean ± SEM. The experiment was repeated twice with similar results.

endothelial cells and immune cells, and moderate signals in intestinal epithelial cells in the lamina propria of the small intestine, after infection, and many p65 signals were localized to the nucleus (Fig 10A). The number of p65-positive cells was partially reduced by anti-TNF-α Ab or α-IL-17A Ab treatment; remarkably, almost all p65 signals were localized to the cytoplasm, suggesting that blockade of TNF-α or IL-17A signaling strongly suppresses transcriptional activations of other genes, such as IL-6, through the NF-κB signaling pathway. By contrast, doxycycline did not prevent p65 nuclear localization (Fig 10A), suggesting that the protective effect of doxycycline was due to blockade of events downstream of NF-κB activation, namely, the direct inhibition to MMP-8 activity. Furthermore, phospho-STAT-3 (Tyr705) was detected in leukocytes and intestinal epithelial cells in the lamina propria of the small intestine of infected mice (Fig 10B), indicating strong signaling through the IL-6R. The phospho-STAT-3 signals were reduced partly by α-TNF-α or α-IL-17A Ab treatment. Additionally, we stained for DENV E protein to identify infected cells in the small intestine. Numerous infiltrating IbaI (a microglia/macrophage marker)-positive macrophages were observed in the lamina propria (Fig 10C), and the major target of DENV was Iba1-positive activated macrophages, not γδ T cells.

## Discussion

The protective effect afforded by anti-TNF-α Ab treatment appeared more evident in the small intestine (Fig 2B and 2G), which suggests the importance of the small intestine in disease pathogenesis. Watanabe *et al.* reported the importance of the small intestine in pathogenesis driven by antibody-dependent enhancement [41]. Our observations may be mouse-specific, but gastrointestinal manifestations, including abdominal pain, abdominal distention, diarrhea, nausea, vomiting, and ascites are common among dengue patients [42, 43]; in particular, gastrointestinal bleeding is a sign of severe dengue or dengue shock syndrome [44, 45]. Furthermore, gastrointestinal manifestations are a common feature in patients with Ebola virus-induced severe hemorrhagic fever [46]. Intestinal pathophysiology may be key to understanding the cytokine storm associated with these diseases.

This study demonstrated that IL-17A is a prominent cytokine involved in the cytokine storm in this model. Guabiraba *et al.* reported that IL-17A contributes to severe disease in DENV-infected mice, and that IL-17A may be downregulated by IL-22 [47]; however, we observed no change in IL-22 levels after anti-TNF-α Ab treatment (accession number GSE199821). IL-17A was instead regulated by TNF-α in this model (Figs 3B, 5G, S1J and S2C). These differences may reflect variations in the virus strains and mouse models employed in

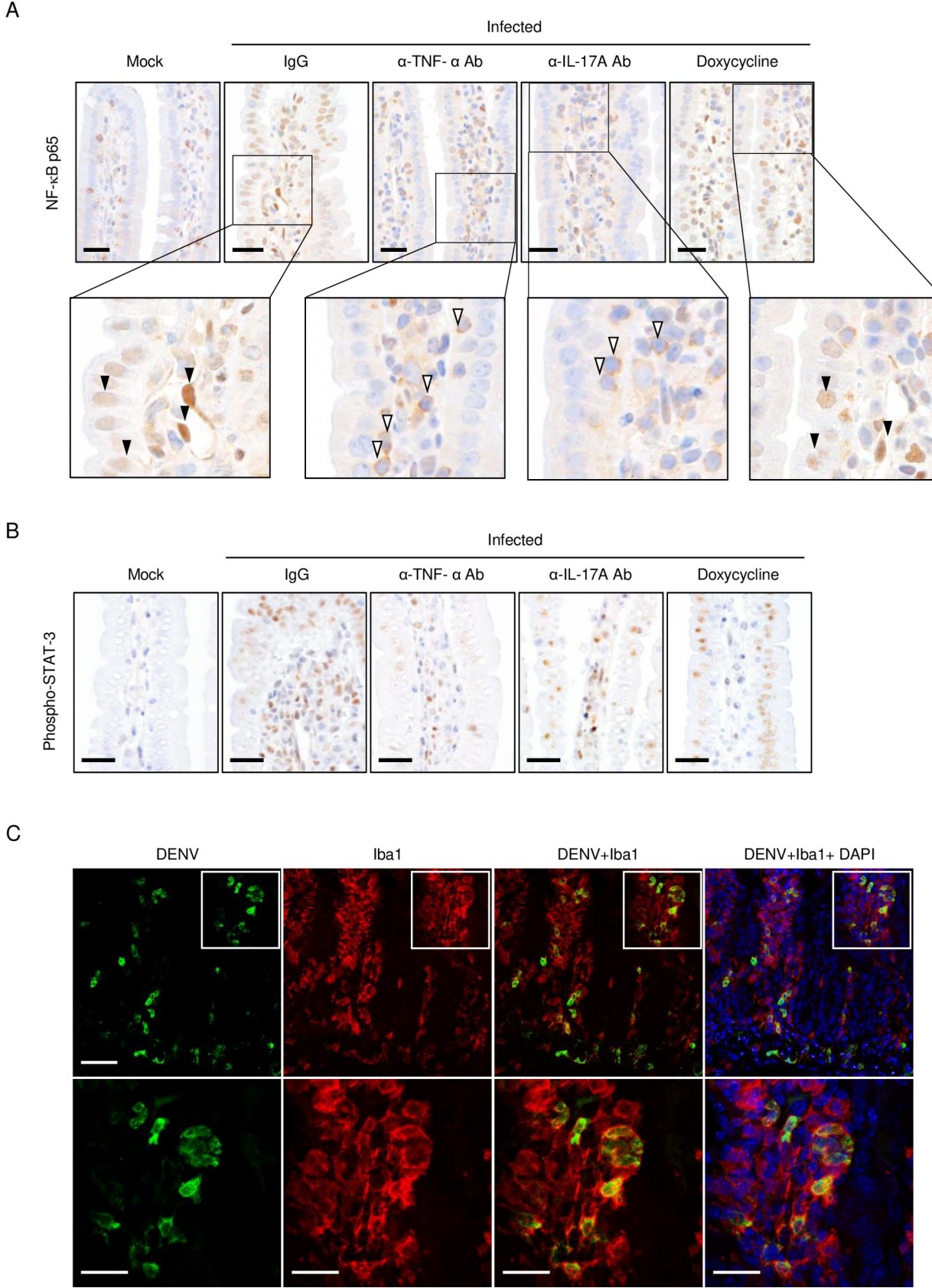

**Fig 10. Detection of NF-κB p65 and phospho-STAT-3 in the small intestine.** IFN-α/β/γR knockout (KO) (8–10 weeks old) mice infected with $2 \times 10^6$ focus-forming units (FFU) of DV3P12/08 or mock-infected mice with PBS were treated with IgG, an α-TNF-α Ab, an α-IL-17A Ab, or doxycycline, and euthanized on Day 4 post-infection. Intestinal sections were subjected to immunostaining with (A) anti-p65 or (B) anti-phospho-STAT-3 Abs. Images are representative of at least three sections from each tissue. The closed arrowheads indicate nuclear-translocated NF-κB p65 and the open arrowheads indicate cytoplasmic NF-κB p65. (C) Confocal laser scanning microscopy shows the localization of DENV E antigen (green), Iba1 (red), and DAPI (blue) in the small intestine of infected IFN-α/β/γRKO mice (8–10 weeks old). The lower panels show high-power images of the insets in the upper panels. Bars = 25 μm.

these studies. IL-17A acts as a potent inducer of systemic inflammation via the induction of inflammatory cytokines (e.g., TNF-α, IL-1β, and IL-6), chemokines, and MMPs [48,49]. Blockade of IL-17A signaling suppressed the induction of major inflammatory cytokines (TNF-α, IL-6, IL-1β, and IL-12) and the anti-inflammatory cytokine IL-10, as well as MMP-8 (Fig 5E and 5F, 5H, 5J and 5L). These effects were similar to the effects of blocking TNF-α signaling (S2 Fig). IL-17A probably enhances these factors by acting synergistically with TNF-α, because blocking one suppressed the expression of the other (Figs 3B, 5E, S1F and S2C). Blockade of TNF-α or IL-17A signaling strongly inhibited nuclear localization of NF-κB p65 (Fig 10A). Suppression of NF-κB activity appears to be the major protective effect of blocking these signaling pathways. As a result, production of IL-6 was inhibited, and phosphorylation of STAT-3 was suppressed (Figs 5F and 10B), thereby preventing vascular leakage. However, it is important to note that TNF-α inhibition increases survival more than IL-17A inhibition (Figs 1F and 5A). In addition, blockade of TNF-α signaling alleviated body weight loss at day 4 p.i. (Fig 1G), whereas blockade of IL-17A signaling did not (Fig 5B). This suggests that TNF-α is not only involved in NF-κB-mediated disease progression, but also in other pathways of disease progression from the onset of the disease. However, IL-17A was upregulated late in the disease (Fig 3B). Therefore, body weight continued to decrease by day 6 p.i. in the surviving mice (Fig 5B). Blockade of IL-6 signaling did not protect mice (Fig 4A) because the level of IL-6 was extremely high in this model (Fig 5F). In contrast, the level of TNF-α or IL-17A was moderate (Fig 5E and 5G). Therefore, inhibiting molecules showing a low or moderate level may be more efficient. However, these findings do not indicate that anti-IL-6 treatment is not appropriate for treating acute infectious diseases. The MR16-1 antibody is the original mouse antibody from which humanized tocilizumab was derived, and is thought to be a good antibody to block the IL-6 signaling. Tocilizumab actually improved mortality in severe Coronavirus Disease 2019 (COVID-19) pneumonia [50]. In human cases of COVID-19 or dengue, IL-6 levels are not as high as those in mice [51]. The possibility of immunotherapy against severe dengue by targeting IL-6 or IL17A should be carefully considered.

We identified γδ T cells as the major IL-17A producers in the small intestine (Fig 6C). Infection induced the selective expansion of Vγ4TCR and Vγ6TCR γδ T cells in the small intestine (Fig 6H). These cells are known to be major IL-17A-producers [29, 52]. These observations raised the question of where the Vγ4 and Vγ6TCR γδ T cells in the small intestine originated from. In mock-infected mice, Vγ4 and Vγ6TCR γδ T cells were found in the liver, thymus, and spleen (Fig 6H). Importantly, a similar pattern of Vγ chain expression was observed in the thymus of infected mice (Fig 6H). It is possible that Vγ4 and Vγ6TCR γδ T cells were recruited from organs, such as the thymus, to the small intestine. When γδ T cells derived from the thymus of mock-infected donor mice and labeled with the intracellular fluorescent label, carboxyfluorescein diacetate succinimidyl ester (CFSE), were transplanted into infected mice, these cells were detected in the small intestine and were undergoing cell division (S5 Fig). Vγ4 or Vγ6 TCR γδ T cells can be recruited from other organs, although these cells can be expanded from small populations of γδ T cells in the small intestine (Fig 6H).

Another important question that was raised was how Vγ4 and Vγ6TCR γδ T cells are activated to produce IL-17A. γδ T cells can be stimulated in a TCR-dependent or -independent

manner [53]. γδTCR can directly recognize stress stimuli, host-cell-derived molecules, and microbial signals in a TCR-dependent manner [54]. For example, γδTCR directly recognizes MHC-related molecules, such as T10/T22 and Qa-1 in mice, MICA/MICB in humans, and CD1d in mice and humans [54]. However, stimulation of γδ T cells in our model appeared not to be TCR-dependent because two different subclasses of γδ T cells were expanded (Fig 6H). A combination of IL-1β and IL-23 alone is sufficient to induce the production of IL-17A by γδ T cells *in vitro* [55]. In our model, the serum levels of IL-1β and IL-23 increased after infection (Figs 5H and S6). IL-1β and IL-23 might be major stimulators in our model. However, this still does not explain why Vγ4 and Vγ6TCR γδ T cells were selectively expanded. Additional stimulation by other surface molecules expressed on Vγ4 and Vγ6TCR γδ T cells strongly activates γδ T cells. Matin *et al.* reported that Toll-like receptor (TLR) 2 and dectin-1 (CLEC7A) induced IL-17A production from γδ T cells [55]. It has been reported that DENV interacts with TLR2 expressed on patient monocytes [56]. γδ T cells are known to express TLR2 [57]. In fact, several TLRs, including TLR2, were upregulated in the small intestine (accession number GSE199821). As IL-17A-producers, Vγ4 and Vγ6TCR γδ T cells are attracting more attention. Vγ4 and Vγ6TCR γδ T cells have been reported to promote the growth of colorectal cancer by producing IL-17A [58]. In addition, *Mycobacterium tuberculosis* is known to stimulate the production of IL-17A by Vγ4 and Vγ6 TCR γδ T cells [59]. In the near future, studies should clarify the detailed mechanism responsible for the stimulation of Vγ4 and Vγ6TCR γδ T cells.

Anti-TNF-α or anti-IL-17A Ab treatment suppressed the infection-induced upregulation of MMP-8 (Figs 3C and 3D, 5L, S1G and S1H). Conversely, doxycycline treatment did not inhibit the production of MMP-8 (Fig 7F and 7G); however, doxycycline treatment preserved the tissue structure of the small intestine (Fig 9B and 9D) and suppressed vascular leakage (Fig 7E). Thus, it is likely that MMP-8 is an important effector of vascular leakage, acting downstream of the cytokine event. MMP-8 is mainly produced by neutrophils [33]. Indeed, large numbers of neutrophils were recruited to the small intestine (Fig 9F). Increased neutrophil counts and a correlation between MMP-8 levels and neutrophilia have been reported in patients with severe dengue [60]; however, MMP-8 has not been reported as an inducer of vascular leakage in other areas of research. MMP-8 may act as a major effector in humans with severe dengue. Known substrates for MMP-8 are collagens, fibronectin, and proteoglycans [61]. One possible scenario is that MMP-8 cleaves the endothelial glycocalyx proteoglycan syndecan-1. In our model, the serum levels of syndecan-1 increased after infection (1.4-fold) (S2R Fig), and this may be the result of glycocalyx degradation. The glycocalyx-like layer has been reported to be destroyed by DENV NS1 alone [62, 63]. Together with NS1, MMP-8 may induce vascular leakage.

Several previous studies reported increased IL-17A levels [15–18] or activation of γδ T cells in patients with severe dengue [64]. A clinical study should be conducted to ascertain whether IL-17A and γδ T cells, as well as other IL-17A-producing cells, neutrophils, and MMP-8, are involved in the pathogenesis of severe dengue in humans. Collectively, the data presented herein provide insight into the pathogenic mechanism underlying the cytokine storm associated with severe dengue and may facilitate the development of novel therapeutic approaches.

## Materials & methods

### Ethics statement

All experiments involving animals were performed in animal biological safety level 2 containment laboratories at the National Institute of Infectious Diseases (NIID) in Japan, and in accordance with the animal experimentation guidelines of the NIID. The protocols were approved by the Institutional Animal Care and Use Committee of the NIID (no. 115064,

118009, and 121003). Trained laboratory personnel performed anesthesia of mice via intraperitoneal injection of a mixture of medetomidine, midazolam, and butorphanol prior to viral injection. Mice were euthanized by exsanguination.

## Virus and cells

The virus strain DV3P12/08, which is derived from patients infected with DENV-3 in Thailand in 2008 [22], was propagated in Vero cells. Virus stocks were stored at -80˚C until use. Vero cells were cultured in Eagle's minimum essential medium (Nacalai Tesque, Kyoto, Japan) supplemented with 10% fetal calf serum.

## Mouse experiments

IFN-α/β/γRKO mice, lacking both type I and type II IFN receptors [22], were bred and housed in ventilated cages, and kept under specific pathogen-free conditions. Male and female mice aged between 8 and 12 weeks were used. Mice were anesthetized by intraperitoneal (i.p.) injection of medetomidine, midazolam, and butorphanol tartrate (final concentrations of 0.3 mg/kg, 4 mg/kg, and 5 mg/kg, respectively) and then challenged intraperitoneally with $2 \times 10^6$ focus-forming units (FFU) of DENV P12/08. Following inoculation, mice were weighed and monitored visually at least once daily by scoring morbidity. Mice were euthanized with isoflurane at the time of sample collection when they became moribund or were expected to die due to difficulty in eating/or drinking.

## Protection assays

Mice were injected intraperitoneally with 100 μg of a purified, functional grade anti-mouse TNF-α antibody (Ab) (clone MP6-XT3; eBioscience), with 500 μg of anti-mouse IL-17A Ab (17F3; Bio X Cell), IL-1β (B122; Bio X Cell), or IL-23 (HRPN; Bio X Cell), or 500 μg of an isotype control Ab (G23-8; Bio X Cell), on Days 1 and 2 p.i. For treatment with doxycycline (TAKARA), mice were injected i.p. with 2.25 mg of doxycycline every day until Day 5 p.i.

## Quantitation of vascular permeability

Vascular leakage was examined by intravascular administration of Evans blue (Sigma-Aldrich), as previously described [22]. Briefly, Evans blue (0.2 ml of a 0.5% solution in PBS) was injected intravenously into moribund mice on Day 4 p.i., along with DENV-3 P12/08. After 2 h, the mice were anesthetized (by i.p. injection of medetomidine, midazolam, and butorphanol tartrate at final concentrations of 0.3 mg/kg, 4 mg/kg, and 5 mg/kg, respectively), euthanized by exsanguination, and perfused extensively with PBS. The liver and small intestine were then collected. Evans blue was extracted from the organs by incubation in 1 ml of formamide (Sigma-Aldrich) for 24 h, followed by centrifugation at $3,000 \times g$ for 10 min; 150 μl of supernatant was collected. The concentration of Evans blue in each organ was quantified by measuring the absorbance at 620 nm using a Corona Grating Microplate Reader SH-9000 (Corona, Electric Co., Ltd). The results were expressed as optical density per gram of tissue.

## Microarray and bioinformatics analyses

Microarray analyses were performed using the following protocol. The liver and small intestine were dissected, and total RNA was extracted using the miRNeasy Extraction Kit (Qiagen). Integrity of total RNA was assessed using an Agilent Bioanalyzer, and the RNA Integrity Number was calculated ($9.0 \geq$ for all samples). A one-color experiment was performed by

hybridizing complementary RNA (cRNA), labeled with either cyanine (Cy)-3 or Cy-5 (Perkin-Elmer), to a Whole Mouse Genome Oligo Microarray 4 × 44 K ver.2 (G4846A; Agilent Technologies). Data were submitted to the NCBI Gene Expression Omnibus under accession number GSE199821. Genes (p < 0.1, two-tailed Student's t-test, paired) showing a mean fold change in expression of > +2.0 or < −2.0 ("+" indicates upregulated and "−" indicates downregulated for each absolute value) in both experiments are listed in S1 and S2 Appendices. Information. Data were analyzed by IPA (Qiagen). Pathway analysis was performed using Gene Set Enrichment Analysis (GSEA) and KEGG pathway gene sets. Analysis was performed by the Integrated Differential Expression and Pathway analysis (iDEP) tool [65] using the following default parameters for pathway analysis: pathway significance, cutoff = 0.2; minimum gene set size = 5. The top 30 significant pathways were obtained.

## Quantitative RT-PCR analyses of host genes and DENV P12/08 RNA

Liver, small intestine, and spleen were homogenized using a TissueLyser II (Qiagen). For cDNA synthesis, total RNA was isolated from the tissue homogenate using TRIzol reagent (Thermo Fisher Scientific). One step, real-time quantitative RT-PCR amplification with SYBR Green I was performed with a Light Cycler (Roche) and the One Step SYBR PrimeScript RT-PCR Kit II (Takara). The final concentration of each PCR primer was 0.08 μM, and the concentration of total RNA was 8 μg/mL, with a reaction volume of 12.5 μL. The conditions for reverse transcription were as follows: 42°C for 5 min, followed by 95°C for 10 sec. PCR amplification used 45 cycles of 95°C for 5 s, 55°C for 30 s, and 72°C for 30 s. For IL-17A and MMP-7, real-time quantitative RT-PCR was performed with the One Step PrimeScript RT-PCR kit (Takara) and FAM-IBFQ probes (Integrated DNA technologies). The conditions for reverse transcription were 42°C for 5 min, followed by 95°C for 10 sec. PCR amplification used 45 cycles of 95°C for 5 s, followed by 60°C for 20 s. To quantify RNA derived from organs, the amounts were normalized to that of total RNA from the corresponding organs from mock-infected mice. Data were analyzed with LightCycler 96 Software ver. 1.1.0.1320 (Roche). The primer sets and probes are listed on S3 Table.

For quantification of viral RNA (vRNA) isolated from the tissue homogenate, total RNA was adjusted to 100 μg/ml for use in real-time PCR. RNA was quantified using a One Step SYBR PrimeScript RT-PCR Kit II (Takara) and the following dengue group-specific primers: DN-F, 5'-CAATATGCTGAAACGCGAGAGAAA-3', and DN-R, 5'-CCCCATCTATTCA-GAATCCCTGCT-3' [22]. The reaction conditions were as follows: 50°C for 30 min, 95°C for 15 min, and then 40 cycles of 95°C for 20 sec, 55°C for 30 sec, and 72°C for 30 sec, followed by a melting curve analysis step. PCR was performed in a LightCycler 96 (Roche). The quantity of vRNA in the initial total RNA was determined by interpolation analysis from a standard curve generated from 10-fold serial dilutions of *in vitro*-transcribed DENV-2 R05-624 RNA made with the MEGAscript Kit (Ambion)[22].

## Quantification of host factors in serum using the Luminex Assay or ELISA

Twenty micro liters of serum were diluted with 80 μL of PBS and the levels of host factors were measured using a Mouse Magnetic Luminex Assay kit (RD systems, Minneapolis, MN, USA). The kits included a 24-plex assay. The mean fluorescence intensity of TNF-α, IL-6, IL-17A, IL-1β, IL-2, IL-10, IL-12p70, IFN-γ, MMP-8, MCP-1, GM-CSF, M-CSF, VEGF, ICAM-1, P-selectin, Syndecan-1, and Podocalyxin was measured by using the Luminex 200 System (Luminex Co., Austin, TX, USA). Total MMP-3 in sera was measured using the Mouse Total MMP-3 ELISA Kit (Proteintech, Rosemont, IL, USA).

## Flow cytometry analysis

The small intestine from the duodenum to the small intestine-cecum junction was removed on Day 4 p.i. Mesenteric fat was removed and the intestine was opened longitudinally, washed in PBS to remove fecal matter, and shaken for 20 min at 37°C in HBSS (WAKO) containing 5 mM EDTA. After removal of epithelial cells and fat tissue, single cell suspensions were prepared by mincing the tissues and incubating them for 1 h at 37°C (with agitation) with 4 mL of RPMI1640 (SIGMA) containing 4% bovine serum albumin (BSA) (WAKO), 1 mg/mL collagenase type 2 (Worthington Biochemical Cooperation), 1 mg/mL dispase II (WAKO), and 40 μg/mL DNase I (Roche). For detection of IL-17A, brefeldin A and monensin (eBioscience) were added to the HBSS buffer and RPMI1640 medium. Digested tissues were filtered through a 70 μm filter and washed with RPMI1640. The resulting cells were pelleted and washed with 20 mL of HBSS supplemented with 5 mM EDTA. Cells were resuspended in 5 mL of 30% Percoll (GE Healthcare), overlayed onto 4 mL of 80% Percoll, and centrifuged at 1,200 × g for 30 min. Isolated lamina propria cells were collected from the interface of the Percoll gradient and washed with RPMI1640. For FACS analysis, cells were resuspended in FACS buffer (HBSS supplemented with 0.5% BSA), incubated at 4°C for 10 min with TruStain FcX anti-mouse CD16/32 antibody (BioLegend), and then stained at 4°C for 30 min with the following fluorochrome-conjugated antibodies: CD45 (30-F11, BD), CD3ε (145-2C11, BioLegend), TCR gamma/delta (GL-3, Invitrogen), IL-17A (TC11-18H10.1, BioLegend), CD11b (M1/70, BioLegend), Ly6G (1A8, BioLegend), Ly6C (HK1.4, BioLegend), and F4/80 (BM8, Invitrogen). After fixation with Cytofix (BD) and permeabilization with Perm/Wash buffer (BD), cells were stained for 1 h at 4°C with an antibody specific for IL-17A (TC11-18H10.1, BioLegend). Stained cells were analyzed with a FACSCalibur (BD) and a FACSLyric (BD) cytometer, and data was analyzed using FlowJo software (BD Biosciences).

## Isolation of γδ T cells and V gene segment usage analysis

Mice were inoculated with $2 \times 10^6$ FF of DENV-3 P12/08. Intestine lamina propria cells and liver cells were harvested on day 4 p.i. and prepared for flow cytometry analysis using the same protocol. For isolation of thymocytes and splenocytes, thymus and spleen were homogenized and filtered through a 70 μm filter, washed with 20 mL of PBS, and the resulting cells were pelleted and washed three times with ice-cold PBS. For the isolation of liver or intestinal cells, single cell suspensions were obtained by mincing the tissues and incubating them for 1 h at 37°C (with agitation) with 4 mL of RPMI1640 (SIGMA) containing 4% bovine serum albumin (BSA) (WAKO), 1 mg/mL collagenase type 2 (Worthington Biochemical Cooperation), 1 mg/mL dispase II (WAKO), and 40 μg/mL DNase I (Roche). Digested tissues were filtered through a 70 μm filter and washed with RPMI1640. The resulting cells were pelleted and washed with 20 mL of HBSS supplemented with 5 mM EDTA. Cells were resuspended in 5.5 mL of HBSS supplemented with 5 mM EDTA, overlayed onto 4 mL of Ficoll-Paque PREMIUM 1.084 (SIGMA) and centrifuged at 700 × g for 40 min. Isolated cells were collected from the interface of the Ficoll gradient and washed with RPMI1640. The γδ T cells were isolated using the TCRγ/δ+ T cell isolation kit (Miltenyi Biotec) according to the following manufacturer's protocol. The V gene segment analysis method was described in a previous paper [30] with minor modification. Total RNA was extracted using TRizol reagent (Thermo Fisher Scientific). RNA was primed with 20 pmol of γ-chain C region (Cγ) primer (5′ – `CTTATGGAGGATTTGTT TCAGC`–3′) for reverse transcription at 30°C for 10 min and 42°C for 60 min with the PrimScript II 1st strand cDNA Synthesis Kit (Takara). PCR was performed on a TaKaRa PCR Thermal Cycler Dice (Takara). PCR cycles were run for 10 s at 98°C, 30 s at 54°C, and 30 s at 72°C; after 35 cycles, extension was prolonged for 4 min at 72°C. The following 5' V primers were

used: Vγ1/2, 5′–ACACAGCTATACATTGGTAC–3′; Vγ2, 5′–CGGCAAAAAACAAATCAAC
AG–3′; Vγ4, 5′–TGTCCTTGCAACCCCTACCC–3′; Vγ5, 5′–TGTGCACTGGTACCAACT
GA–3′; Vγ6, 5′–GGAATTCAAAAGAAAACATTGTCT–3′; and Vγ7, 5′–AAGCTAGAGGGG
TCCTCTGC–3′. PCR products (10 μl) were subjected to electrophoresis on a 1.5% agarose gel
(Iwai Chemicals Company) and then transferred to Hybond-N (GE Healthcare). Southern
blots of the γ PCR products were hybridized with a digoxigenin (DIG)-labeled Jγ1 probe (5′–
AGAGGGAATTACTATGAGCT–3′). Before hybridization, the filters were incubated for 20
min in Perfect Hybridization Solution (PHS) (TOYOBO), and then hybridized for 2 h at 48˚C
with the probe in PHS. The filters were washed twice, each for 5 min at 48˚C, with 1ˢᵗ wash
buffer (2×SSC containing 0.1% SDS), followed by two washes in 0.5×SSC containing 0.1% SDS
for 15 min at 48˚C twice. For DIG detection, the membrane was incubated with blocking solu-
tion, incubated with an alkaline phosphatase-labeled anti-DIG antibody, washed with washing
buffer using the DIG Wash and Block Buffer Set (Roche), and then incubated with disodium
3-(4-methoxyspiro {1,2-dioxetane-3,2′-(5′-chloro)tricyclo [3.3.1.13,7]decan}-4-yl)phenyl
phosphate (CSPD, ready-to-use) (Roche). Signals were detected using a LAS-3000 apparatus
(Fujifilm).

## Histopathology and immunohistochemistry

Mice were anesthetized and perfused with 10 ml of 10% phosphate-buffered formalin. Next,
the liver and small intestine were harvested and fixed. Fixed tissues were embedded in paraffin,
sectioned, and stained with hematoxylin and eosin (H&E). For Ly6b staining, no antigen
retrieval was performed. Then, endogenous peroxidase was quenched by immersing sections
for 10 min in 10% $H_2O_2$/PBS. After washing with PBS and blocking for 30 min at room tem-
perature with 5% BSA, sections were incubated with anti-Ly6b antibody (1:200) overnight at
4˚C. Neutrophils were identified by staining with an anti-Ly6b Ab. For immunohistochemis-
try, antigen (Claudin-3 and NF-κB) retrieval was performed by autoclaving sections at 121˚C
for 10 min in target retrieval solution (pH 6.0; Nichirei, Tokyo, Japan) or target retrieval solu-
tion pH 9.0 (Nichirei) (for phosphor STAT-3). For Ly6b staining, no antigen retrieval was per-
formed. Then, endogenous peroxidase was quenched by immersing sections for 10 min in
10% $H_2O_2$/PBS. After washing with PBS and blocking for 30 min at room temperature with
5% BSA, sections were incubated overnight at 4˚C with an anti-Claudin-3 antibody (1:200,
Thermo Fisher Scientific), an anti-NF-κB antibody (clone D14E12; 1:500, Cell Signaling Tech-
nology), an anti-phospho-STAT3 antibody (clone D3A7, 1:300, Cell Signaling Technology), or
an anti-Ly6b antibody (1:200, Ly-6B.2 alloantigen antibody 7/4; Bio-Rad). After washing, sec-
tions were incubated with Histofine simple stain MAX-PO anti-rabbit secondary antibody
(Nichirei) or Histofine simple stain MAX-PO anti-rat secondary antibody (Nichirei). Signals
were visualized with Histofine simple stain DAB solution (Nichirei) and counter staining was
performed with hematoxylin. The resulting sections were examined under a BX-53 micro-
scope (Olympus). For immunofluorescence staining of DENV E antigen and Iba1 in the small
intestine of infected IFN-α/β/γRKO mice, the small intestine was fixed with 4% paraformalde-
hyde and cut into small pieces to make cryosections. The tissue was dehydrated by sequential
immersion overnight at 4˚C in 10% and 30% sucrose solutions. Then, the sections were
embedded in OCT compound (Sakura Finetek, Tokyo, Japan) and stored at -80˚C. Next,
10 μm tissue sections were prepared using a Cryostat CM1900 (Leica Biosystems, Wetzlar,
Germany). The resulting sections were washed in water and PBS, blocked with 5% BSA, and
incubated at 4˚C overnight with mixture of human monoclonal anti-DENV E protein (clone;
D23-1G7C21:400)[66] and a rabbit polyclonal anti-Iba1 antibody (1:1000, Fujifilm-Wako).
After washing with PBS, the sections were incubated for 40 min at room temperature with

Alexa488-conjugated anti-human IgG (1:400, ThermoFisher Scientific) and Alexa594-conjugated anti-rabbit IgG (1:400, Thermo Fisher Scientific). After washing, the sections were mounted with Vectashield mounting medium (hard set) containing DAPI (Vector Laboratories, Burlingame, CA, USA). The fluorescent signals were visualized under a confocal microscope (Fluoview FV3000; Olympus, Tokyo, Japan).

## Data analysis

In all the bar graphs and scatter plots, data are expressed as mean ± standard error of the mean (SEM). All data were analyzed using GraphPad Prism software (GraphPad, San Diego, CA, USA). Statistical analysis was performed by one-way ANOVA and the significance of differences was assessed by Tukey's multiple comparison test for multiple comparisons. Comparison between two groups was performed using two-tailed Student's $t$-test. For the graphs for IL-6 mRNA, IL-17A mRNA, MMP-8, MMP-3 mRNA, vRNA, IL-6, IL-17A, IL-10, IFN-γ, MMP-8, and MMP-3, log transformed data were used for statistical analysis. Survival data were evaluated by the Log-rank (Mantel-Cox) test. A $p$ value of $< 0.05$ was considered statistically significant; $*P < 0.05$, $**P < 0.01$, $***P < 0.001$, and $****P < 0.0001$. Details of sample sizes are included in all figure legends.

## Supporting information

**S1 Fig. Time course of host mRNA expression.** Levels of mRNA expression of TNF-α (A and B), IL-6 (C and D), IL-17A (E and F), MMP-8 (G and H), MMP-7 (I and J), and MMP-3 (K and L) in the liver (A, C, E, G, I, and K) and small intestine (B, D, F, H, J, and L) of mock- (8–10 weeks old) and DENV-3 P12/08 ($2 \times 10^6$ focus-forming units [FFU])-infected mice. The time points on the $x$-axis refer to days post-infection. Total RNA was extracted from the liver (n = 4–8) or small intestine (n = 4–8) and subjected to quantitative RT-PCR. The expression of each mRNA was calculated relative to that in mock-infected mice. Viral titers in the liver (M), small intestine (N), and spleen (O) were measured by quantitative RT-PCR. The results are expressed as the mean ± SEM. Each symbol represents an individual mouse. Data were analyzed by one-way ANOVA and significance was assessed by Tukey's multiple comparison test. $*P < 0.05$, $**P < 0.01$, $***P < 0.001$, and $****P < 0.0001$.
(TIF)

**S2 Fig. Serum host factor concentrations.** IFN-α/β/γR knockout (KO) mice (8–10 weeks old) were intraperitoneally infected with $2 \times 10^6$ focus-forming units (FFU) of DENV-3 P12/08. Isotype control IgG (n = 4), anti-IL-17A Ab (n = 12), or anti-TNF-α Ab (n = 4) was inoculated intraperitoneally at Days 1 and 2 post-infection (p.i.). The mice were sacrificed under anesthesia at Day 4 p.i. and sera were collected. Concentrations of TNF-α (A), IL-6 (B), IL-17A (C), MMP-8 (D), IL-12p70 (E), MCP-1 (F), IL-10 (G), IL-1β (H), IFN-γ (I), IL-2 (J), M-CSF (K), GM-CSF (L), VEGF (M), ICAM-1 (N), P-selectin (O), and syndecan-1 (P) in sera were determined by the Mouse Magnetic Luminex Assay kit. Concentrations of all factors were analyzed by one-way ANOVA. Significance was assessed by Tukey's multiple comparison test. $*P < 0.05$, $**P < 0.01$, $***P < 0.001$ and $****P < 0.0001$. ns indicates no significant difference.
(TIF)

**S3 Fig. Viral titers in the liver, small intestine and spleen.** IFN-α/β/γR knockout (KO) mice (8–10 weeks old) were infected with $2.0 \times 10^6$ focus-forming units (FFU) of DENV-3 P12/08. Anti-IL-17A Ab (n = 4) or isotype control IgG (n = 6) was injected intraperitoneally at Days 1 and 2 post-infection, and the mice were sacrificed at Day 4 post-infection. Viral copy numbers in the liver, intestine, and serum were determined by quantitative RT-PCR. Differences in

viral titers were analyzed by Student's *t*-test. ns indicates no significant difference ($P > 0.05$). The results are expressed as the mean ± SEM.
(TIF)

**S4 Fig. Administration of antibiotics in mice.** IFN-α/β/γR knockout (KO) mice (8 weeks old) were administered a combination of four antibiotics, consisting of 1 mg/ml ampicillin (Nacalai Tesque), 1 mg/ml neomycin (Nacalai Tesque), 1 mg/ml metronidazole (Nacalai Tesque), and 500 μg/ml vancomycin (Nacalai Tesque) in sterilized drinking water for 2 weeks. IFN-α/β/γRKO mice were intraperitoneally infected with $2 \times 10^6$ focus-forming units (FFU) of DENV-3 P12/08 (n = 3/group), and the mice were observed up until Day 5 post-infection. Kaplan–Meier survival curves show the percentage of mice surviving on the specified days post-infection. Significant differences between individual groups were evaluated using the log-rank (Mantel–Cox) test. $P > 0.9999$.
(TIF)

**S5 Fig. Analysis of CFSE-labeled transfer cells.** Representative flow cytometry plots of gated intestinal cells. IFN-α/β/γR knockout (KO) mice (8–10 weeks old; n = 3/group) were intraperitoneally infected with $2 \times 10^6$ focus-forming units (FFU) of DENV-3 P12/08 or mock. CFSE-labeled cells from the thymus or the intestine were intravenously transferred, and the mice were sacrificed under anesthesia at Day 4 post-infection. The small intestine was collected. (A) The SSC-FSC profile was used to distinguish leukocyte populations from other cell populations. (B) Duplets were removed by singlet gating and debris was removed from the analysis. (C) CFSE-positive cells were further gated. (D) γδ T cells were gated by γδTCR-CD45 analysis. (E) CFSE-labeled input cells. (F) Thymus-derived CFSE-labeled γδ T cells isolated from the small intestine of infected IFN-α/β/γRKO mice.
(TIF)

**S6 Fig. Serum IL-23 concentrations.** IFN-α/β/γR knockout (KO) mice (8–10 weeks old) were intraperitoneally infected with $2 \times 10^6$ focus-forming units (FFU) of DENV-3 P12/08 or mock (n = 9). Isotype control IgG (n = 5), anti-TNF-α (n = 4), or anti-IL-17A Ab (n = 3) was intraperitoneally inoculated at Days 1 and 2 post-infection, and the mice were sacrificed under anesthesia at Day 4 post-infection. Sera were then collected. Serum IL-23 concentrations were determined by the Mouse IL-23 Quantikine ELISA Kit (R&D systems). IL-23 concentrations were analyzed by one-way ANOVA. Significance was assessed using Tukey's multiple comparison test. $^*P < 0.05$, $^{**}P < 0.01$, $^{***}P < 0.001$, and $^{****}P < 0.0001$. ns indicates no significant difference.
(TIF)

**S1 Table. Upstream regulator analysis.**
(DOCX)

**S2 Table. Top analysis-ready molecules.**
(DOCX)

**S3 Table. Real-time PCR primer and probe sequences.**
(DOCX)

**S1 Appendix. Microarray analysis in the liver.**
(CSV)

**S2 Appendix. Microarray analysis in the small intestine.**
(CSV)

## Acknowledgments

The authors would like to thank Chugai Pharmaceutical Co., Ltd for providing MR16-1. We thank Edanz (https://jp.edanz.com/ac) for editing a draft of this manuscript.

## Author Contributions

**Conceptualization:** Takeshi Kurosu, Daisuke Okuzaki, Yusuke Sakai, Mohamad Al Kadi, Supranee Phanthanawiboon, Yasusi Ami, Masayuki Shimojima, Tomoki Yoshikawa, Shuetsu Fukushi, Noriyo Nagata, Tadaki Suzuki, Daisuke Kamimura, Masaaki Murakami, Hideki Ebihara, Masayuki Saijo.

**Data curation:** Takeshi Kurosu, Daisuke Okuzaki, Yusuke Sakai, Mohamad Al Kadi, Supranee Phanthanawiboon, Yasusi Ami, Masayuki Shimojima, Tomoki Yoshikawa, Shuetsu Fukushi, Noriyo Nagata, Tadaki Suzuki, Masaaki Murakami, Masayuki Saijo.

**Formal analysis:** Takeshi Kurosu, Daisuke Okuzaki, Yusuke Sakai, Mohamad Al Kadi.

**Funding acquisition:** Takeshi Kurosu.

**Investigation:** Takeshi Kurosu, Daisuke Okuzaki, Yusuke Sakai, Mohamad Al Kadi, Supranee Phanthanawiboon, Daisuke Kamimura.

**Methodology:** Takeshi Kurosu, Daisuke Okuzaki, Yusuke Sakai, Mohamad Al Kadi, Yasusi Ami, Masaaki Murakami.

**Project administration:** Takeshi Kurosu, Daisuke Okuzaki, Yusuke Sakai, Mohamad Al Kadi, Tadaki Suzuki, Masayuki Saijo.

**Resources:** Takeshi Kurosu, Daisuke Okuzaki, Noriyo Nagata, Tadaki Suzuki, Masaaki Murakami, Hideki Ebihara, Masayuki Saijo.

**Software:** Takeshi Kurosu, Daisuke Okuzaki, Mohamad Al Kadi.

**Supervision:** Takeshi Kurosu, Daisuke Okuzaki.

**Validation:** Takeshi Kurosu, Daisuke Okuzaki, Yusuke Sakai, Mohamad Al Kadi, Supranee Phanthanawiboon.

**Visualization:** Takeshi Kurosu, Daisuke Okuzaki, Yusuke Sakai, Mohamad Al Kadi.

**Writing – original draft:** Takeshi Kurosu, Daisuke Okuzaki, Yusuke Sakai, Mohamad Al Kadi.

**Writing – review & editing:** Takeshi Kurosu, Daisuke Okuzaki, Yusuke Sakai, Mohamad Al Kadi, Supranee Phanthanawiboon, Yasusi Ami, Masayuki Shimojima, Tomoki Yoshikawa, Shuetsu Fukushi, Noriyo Nagata, Tadaki Suzuki, Daisuke Kamimura, Masaaki Murakami, Hideki Ebihara, Masayuki Saijo.

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
