## [Decision Letter · Decision Letter 0]

13 Aug 2023

Dear Dr. Kurosu,

Thank you very much for submitting your manuscript "Dengue virus infection induces selective expansion of Vγ4 and Vγ6TCR γδ T cells in the small intestine and a cytokine storm driving vascular leakage in mice" for consideration at PLOS Neglected Tropical Diseases. As with all papers reviewed by the journal, your manuscript was reviewed by members of the editorial board and by several independent reviewers. In light of the reviews (below this email), we would like to invite the resubmission of a significantly-revised version that takes into account the reviewers' comments. 

The manuscript was evaluated by two experts in the area which appreciated the importance and novelty of the manuscript. The reviewers raised important points regarding to the gating of their population analysis and requested further clarification on the animal models, and protocol and effects of treatments utilized assess cytokine effects on the model, that must be addressed by the authors. The authors also need to mention if and how many times experiments were repeated and attest the reproducibility of results in their hands.

We cannot make any decision about publication until we have seen the revised manuscript and your response to the reviewers' comments. Your revised manuscript is also likely to be sent to reviewers for further evaluation.

Sincerely,

Helton C. Santiago, M.D., Ph.D

Academic Editor

Aaron Jex

Section Editor

The manuscript was evaluated by two experts in the area which appreciated the importance and novelty of the manuscript. The reviewers raised important points regarding to the gating of their population analysis and requested further clarification on the animal models, and protocol and effects of treatments utilized assess cytokine effects on the model, that must be addressed by the authors. The authors also need to mention if and how many times experiments were repeated and attest the reproducibility of results in their hands.

Reviewer's Responses to Questions

**Key Review Criteria Required for Acceptance?**

**Methods**

-Are the objectives of the study clearly articulated with a clear testable hypothesis stated?

-Is the study design appropriate to address the stated objectives?

-Is the population clearly described and appropriate for the hypothesis being tested?

-Is the sample size sufficient to ensure adequate power to address the hypothesis being tested?

-Were correct statistical analysis used to support conclusions?

-Are there concerns about ethical or regulatory requirements being met?

Reviewer #1: Clearly written

Reviewer #2: (No Response)

**Results**

-Does the analysis presented match the analysis plan?

-Are the results clearly and completely presented?

-Are the figures (Tables, Images) of sufficient quality for clarity?

Reviewer #1: Clearly written

Reviewer #2: (No Response)

**Conclusions**

-Are the conclusions supported by the data presented?

-Are the limitations of analysis clearly described?

-Do the authors discuss how these data can be helpful to advance our understanding of the topic under study?

-Is public health relevance addressed?

Reviewer #1: Clearly written

Reviewer #2: (No Response)

**Editorial and Data Presentation Modifications?**

Reviewer #1: (No Response)

Reviewer #2: (No Response)

**Summary and General Comments**

Reviewer #1: In this study, Kurosu et al showed that a selective expansion of IL-17A-producing γδ T cells in the small intestine is at least partially responsible for the induction of lethal vascular leak caused by dengue virus (DENV) in IFN-α/β/γR KO mice. The critical role of IL-17A in causing severe vascular leak was demonstrated with the 50% protection of mouse survival by the treatment with anti-IL-17A antibodies accompanied with the reduction of cytokine gene expressions, such as TNF-α, in small intestine. In addition, the authors showed that DENV infection causes the infiltration of neutrophils in small intestine, resulting in a massive production of MMP-8, which is also shown to be responsible for the induction of vascular leak. Thus, this study presented novel mechanisms of vascular leak caused by DENV infection in IFN-α/β/γR KO mice; (1) IL-17A produced by γδ T cells and (2) MMP-8 produced by neutrophils. However, I would request more convincing data/explanations to accept the author’s claims.

1. Figure 6 is important data to identify the IL-17A-expressing lymphocytes. The gate was first set on CD45-positive and SSC (A). However, the non-lymphocyte population in infected mice seems totally shifted toward the right side of x-axis and the gate included a lot of non-lymphocyte cells. This arises the possibility that the CD3+γδTCR+ population (B) includes a number of non-specific cells. I understand that the activated lymphocytes may shift their positions to higher SSC intensity, but they should be still distributed within the elongated pattern of lymphocyte population. I urge the authors to re-analyze the data with appropriate gating to obtain more convincing results.

2. The treatment with anti-IL-17A antibodies showed 50% protective efficacy (Fig.5 A). This is unconvincing to me since the treat mice showed similar pattern of weight reduction to untreated mice (Fig.5B) and similar levels of cytokine production to anti-TNF antibody treatment (Fig. 5 E-M), which showed 100% protective efficacy (Fig. 1F) and induced less weight loss (Fig. 1G). How do the authors explain this discrepancy?

Reviewer #2: PNTD-D-23-00821

Takeshi Kurosu and colleagues submitted a manuscript entitled “Dengue virus infection induces selective expansion of V�4 and V�6TCR �� T cells in the small intestine and a cytokine storm driving vascular leakage in mice”. In the manuscript, the authors examined the mechanisms and effects of blockade of TNF-� signaling on DENV infected mice during severe dengue outcome. Using a well-established mouse model of severe dengue which display human symptoms including vascular leakage, the authors performed transcriptomic analysis of the liver and small intestine samples collected chronologically in presence/absence of blockade of TNF-� signaling and evaluated the cytokine and effector level events. They found that (i) blockade of TNF-� signaling has a better effect in pro-inflammatory cytokines in small intestine compared to liver, (ii) a critical role of IL-17A produced by �� T cells in the small intestine during severe dengue, and (iii) detected the presence of neutrophil infiltrate that produced MMP-8 (Matrix Metalloproteinase 8), a collagen cleaving enzyme which is a major player of inflammatory response.

The manuscript addresses a critical question in the field regarding the mechanisms of action following severe dengue. The manuscript uses well-defined approaches to explore the questions and discuss the conclusions. The manuscript is well written, however some concerns detailed below should be addressed by the authors to improve the understanding and the quality of the manuscript.

Minor comments:

Figure 1: Panel E needs more details as done for panel A. The symbols are confusing. When is the infection and what antibody has been used?

The authors mentioned that the mice are dying at day 4 in this mouse model, but Fig 1F showed day 5. It is not clear if the mice are dying at day 4 or 5. 

Typos in line 962: Mice were instead of Mice. Were

Figure S2: The effects of TNF-� blockade are mentioned as limited, why? Even though in panel F, the protection is 100%. This is not clear. Please, clarify.

Typos in line 254-255 and 257: The ��TCR subsets seem mislabeled. i.e CD3+/��TCR- (including CD8 T cells) in Fig6D instead of CD3+/��TCR+ (including CD8 T cells). Please, double check, Figure 6C-G

Figure S5: Add the title of the Y axis.

Figure 8: The gating strategy will be better and easy to follow as one panel instead of 5 panels.

Figure 9: The resolution of the image is not great and the authors should add annotations and labels to indicate what histopathological changes should be seen and compared between treatments.

Moderate comments: 

In all figures, the authors did not specify how many times the experiments were done. Moreover, in Figure 1, only 3 to 4 mice have been used. This looks like a one-time experiment and knowing the variability of the mouse experiments, it is not acceptable to do an experiment only once. Please, clarify and/or add more samples to confirm the results.

Figure 4: It is not clear why the authors are not using a better protocol to blockade the IL-6 signaling. The antibody treatment did not effectively suppress IL-6 signaling as measured by serum albumin A levels (Fig 4C). Is it the result of a suboptimal protocol or a bad antibody lot? The use of MR16-1 to suppress IL-6 signaling seems to have better effects in others published manuscripts. The purpose of this figure is not clear if the treatment is not working.

In addition, this mouse model is supposed to succumb to DENV infection at day 4, however this is not the case in panel A. Could the authors clarify why this mouse model is different here?

PLOS authors have the option to publish the peer review history of their article (what does this mean?). If published, this will include your full peer review and any attached files.

Reviewer #1: No

Reviewer #2: No
---

## [Decision Letter · Decision Letter 1]

24 Sep 2023

Dear Dr. Kurosu,

Thank you very much for submitting your manuscript "Dengue virus infection induces selective expansion of Vγ4 and Vγ6TCR γδ T cells in the small intestine and a cytokine storm driving vascular leakage in mice" for consideration at PLOS Neglected Tropical Diseases. As with all papers reviewed by the journal, your manuscript was reviewed by members of the editorial board and by several independent reviewers. The reviewers appreciated the attention to an important topic. Based on the reviews, we are likely to accept this manuscript for publication, providing that you modify the manuscript according to the review recommendations. 

The reviewers acknowledge the improvement in the manuscript, however there is still a minor point that was still not properly addressed. Reviewer number one points that there may be unspecific IL-17+ population based on the gating strategy of the authors in Figure 6. Please, try to respond this concern properly or describe the limitations that this strategy may bring.

Sincerely,

Helton C. Santiago, M.D., Ph.D

Academic Editor

Aaron Jex

Section Editor

The reviewers acknowledge the improvement in the manuscript, however there is still a minor point that was not properly addressed. Reviewer number one points that there may be unspecific IL-17+ population based on the gating strategy of the authors in Figure 6. Please, try to respond this concern properly or describe the limitations that the current strategy may bring.

Reviewer's Responses to Questions

**Key Review Criteria Required for Acceptance?**

**Methods**

-Are the objectives of the study clearly articulated with a clear testable hypothesis stated?

-Is the study design appropriate to address the stated objectives?

-Is the population clearly described and appropriate for the hypothesis being tested?

-Is the sample size sufficient to ensure adequate power to address the hypothesis being tested?

-Were correct statistical analysis used to support conclusions?

-Are there concerns about ethical or regulatory requirements being met?

Reviewer #1: I would request the Authors for an alternative FCM analysis.

Reviewer #2: (No Response)

**Results**

-Does the analysis presented match the analysis plan?

-Are the results clearly and completely presented?

-Are the figures (Tables, Images) of sufficient quality for clarity?

Reviewer #1: Results are described clearly

Reviewer #2: (No Response)

**Conclusions**

-Are the conclusions supported by the data presented?

-Are the limitations of analysis clearly described?

-Do the authors discuss how these data can be helpful to advance our understanding of the topic under study?

-Is public health relevance addressed?

Reviewer #1: Conclusions are reasonable

Reviewer #2: (No Response)

**Editorial and Data Presentation Modifications?**

Reviewer #1: No

Reviewer #2: (No Response)

**Summary and General Comments**

Reviewer #1: Kurosu et al responded to 2 concerns raised by this Reviewer. Overall, their responses are reasonable. One point they missed to answer regarding the FCM data is the possibility that CD3+γδTCR+ population (Fig. 6B) contains a number of non-specific cells by gating a broad range of SSC intensity (Fig. 6A) and this leads the false positive of IL-17A expression (Fig. 6C). I understand that CD45+/SSClow population is mainly lymphocytes and CD45+/SSChigh population would be mainly neutrophils and monocytes. Gating and analyzing separately these populations would be the best way in order to avoid the concern about false-positive and clearly show that γδ T cells are responsible for IL-17A production in small intestine. It would be very easy to address this point.

Reviewer #2: The authors addressed some of the comments and improved the manuscript. Typos and grammatical mistakes are still in the manuscript, ie L1065, L1150, etc...

PLOS authors have the option to publish the peer review history of their article (what does this mean?). If published, this will include your full peer review and any attached files.

Reviewer #1: No

Reviewer #2: No

Figure Files:

Data Requirements:

Reproducibility:

References

---

## [Editor Report · Decision Letter 2]

19 Oct 2023

Dear Dr. Kurosu,

We are pleased to inform you that your manuscript 'Dengue virus infection induces selective expansion of Vγ4 and Vγ6TCR γδ T cells in the small intestine and a cytokine storm driving vascular leakage in mice' has been provisionally accepted for publication in PLOS Neglected Tropical Diseases.

Best regards,

Helton C. Santiago, M.D., Ph.D

Academic Editor

Aaron Jex

Section Editor

Although the authors did NOT answer the reviewer #1 properly, which is not a good practice, we agree that there is not major reasons of concern with figure 6 and the claims of the manuscript.

---

## [Editor Report · Acceptance letter]

31 Oct 2023

Dear Dr. Kurosu,

We are delighted to inform you that your manuscript, "Dengue virus infection induces selective expansion of Vγ4 and Vγ6TCR γδ T cells in the small intestine and a cytokine storm driving vascular leakage in mice," has been formally accepted for publication in PLOS Neglected Tropical Diseases.

Best regards,

Shaden Kamhawi

co-Editor-in-Chief

Paul Brindley

co-Editor-in-Chief
